# Linear algebra with transformers

**François Charton Meta AI**
fcharton@meta.com

**Reviewed on OpenReview:** https://openreview.net/forum?id=Hp4g7FAXXG

## Abstract

Transformers can learn to perform numerical computations from examples only. I study nine problems of linear algebra, from basic matrix operations to eigenvalue decomposition and inversion, and introduce and discuss four encoding schemes to represent real numbers. On all problems, transformers trained on sets of random matrices achieve high accuracies (over 90%). The models are robust to noise, and can generalize out of their training distribution. In particular, models trained to predict Laplace-distributed eigenvalues generalize to different classes of matrices: Wigner matrices or matrices with positive eigenvalues. The reverse is not true.

## 1 Introduction

Since their introduction for machine translation by Vaswani et al. (2017), transformers were applied to a wide range of problems, from text generation (Radford et al., 2018; 2019) to image processing (Carion et al., 2020) and speech recognition (Dong et al., 2018), where they now achieve state-of-the-art performance (Dosovitskiy et al., 2021; Wang et al., 2020b). Transformers have also been proposed for problems of symbolic mathematics, like integration (Lample & Charton, 2019), theorem proving (Polu & Sutskever, 2020), formal logic (Hahn et al., 2021), SAT solving (Shi et al., 2021), symbolic regression (Biggio et al., 2021) and dynamical systems Charton et al. (2020). In these works, transformers perform symbolic computations, i.e. manipulate abstract mathematical symbols.

Beyond symbol manipulation, mathematics also involve numerical calculations (e.g. arithmetic, numerical solutions of equations). On these tasks, experiments with transformers and other sequence models have been disappointing. Basic arithmetic operations, like multiplication or modulus, prove very difficult to learn (Kaiser & Sutskever, 2015; Palamas, 2017), and models struggle with generalization out of their training distribution (Nogueira et al., 2021). It could even be shown (Shalev-Shwartz et al., 2017) that some arithmetic tasks cannot be solved using gradient descent. Such results might severely restrict the applicability of transformers in science. Most practical problems of mathematics mix symbolic and numerical computations. If transformers "cannot compute", their use in science is very limited.

In this paper, I investigate the capability of transformers to learn to perform numerical computations with high accuracy. I focus on nine problems of linear algebra, from basic operations on dense matrices to inversion, eigen and singular value decomposition. I show that small transformers can be trained, from examples only, to compute approximate solutions (up to a few percents of the $L^1$ norm) with more than 90% accuracy (over 99% in most cases). I propose and discuss four encodings to represent real numbers, and train small sequence to sequence transformers (up to 6 layers, 10 to 50 million trainable parameters) from generated datasets of random matrices. I investigate different architectures, in particular asymmetric configurations where the encoder or decoder has only one layer. Finally, I show that the models are robust to noisy data, and that they can generalize out of their training distribution if special attention is paid to training data generation.

**Caveat.** This paper does not advocate replacing existing linear algebra algorithms with transformer-based implementations. Numerical packages are faster, more accurate, and scale better. My motivation is twofold: better understand the capabilities and limitations of transformers in mathematics, and investigate their potential use as tools for the emerging field of AI for science.

In applications to mathematics, while previous research has shown that transformers struggle with basic arithmetic, I demonstrate that they can learn complex computations, like eigenvalue decomposition. I also show that leveraging the theory of random matrices can help understand the mechanisms of out-of-domain generalization, a known limitation of transformers in mathematics (Welleck et al., 2021), and a difficult problem because of the lack of metrics over problem space.

Beyond mathematics, transformers are fast becoming the "default model" for many deep learning applications. Their potential use as end to end tools for AI for Science has received a lot of attention recently. I believe that demonstrating that transformers can handle some of the computational building blocks of many scientific problems, like the problems of linear algebra discussed here, is a pre-requisite to their wider generalization.

The source code for the model and experiments is available at `github.com/facebookresearch/LAWT`.

## 2 Problems and datasets

Let $M$ and $N$ be $m \times n$ real matrices and $V \in \mathbb{R}^m$ . This paper considers nine problems of linear algebra:

- matrix transposition: find $M^T$, a $n \times m$ matrix,
- matrix addition: find $M + N$, a $m \times n$ matrix,
- matrix-vector multiplication: find $M^T V$, in $\mathbb{R}^n$,
- matrix multiplication: find $M^T N$, a $n \times n$ matrix,
- eigenvalues: $M$ symmetric, find its $n$ (real) eigenvalues, sorted in descending order,
- eigenvectors: $M$ symmetric, find $D$ diagonal and $Q$ orthogonal such that $QMQ^T = D$, set as a $(n+1) \times n$ matrix, with (sorted) eigenvalues in its first row,
- singular values: find the $n$ eigenvalues of $M^T M$, sorted in descending order,
- singular value decomposition: find orthogonal $U, V$ and diagonal $S$ such that $S = UMV$, set as a $(m + n + 1) \times min(m, n)$ matrix,
- inversion: $M$ square and invertible, find its inverse $P$, such that $MP = PM = Id$.

These problems range from operations on single coefficients of the matrices (transposition and addition), to computations over rows and columns, involving several arithmetic operations (multiplication), and complex nonlinear transformations involving the whole matrix (decompositions and inversion).

For each problem, the training data is generated by sampling random input matrices $I$ (see section 2.2), and computing the output $O$ with a linear algebra package (NumPy linalg). All coefficients in $I$ and $O$ are set in base ten floating-point representation, and rounded to three significant digits in the mantissa. If a problem has several input or output matrices, they are concatenated into one (for instance, the two $m \times n$ operands of the addition task are concatenated into one $m \times 2n$ matrix $I$).

### 2.1 Encoding matrices as sequences

The input and output to all problems studied here are matrices. Transformers process sequences of tokens. To encode a $m \times n$ matrix as a sequence, its dimensions are encoded as two symbolic tokens (`Vm` and `Vn`), and its $mn$ coefficients are then enumerated and encoded. I propose four encoding schemes for matrix coefficients (set in scientific notation with three significant digits): P10, P1000, B1999, and FP15.

**Base 10 positional encoding (P10)** represents numbers as sequences of five tokens : one sign token (`+` or `-`), 3 digits (from `0` to `9`) for the mantissa, and a symbolic token (from `E-100` to `E+100`) for the exponent. For instance, 3.14 is represented as $314.10^{-2}$, and encoded as [`+`, `3`, `1`, `4`, `E-2`].

**Base 1000 positional encoding (P1000)** provides a more compact representation. The mantissa is encoded as a single token (from `0` to `999`) and a number is represented as the triplet (sign, mantissa, exponent).

**Balanced base 1999 (B1999)** encodes the sign and mantissa as a single token (from `-999` to `999`).

**15 bit floating point (FP15)** encodes a floating point number $x = m10^b$ as a single token `FPm/b`.

Table 1 provides examples for the four encodings. More information can be found in Appendix A.

| Encoding | 3.14 | $-6.02.10^{23}$ | Tokens / coefficient | Size of vocabulary |
|---|---|---|---|---|
| P10 | [+, 3, 1, 4, E-2] | [-, 6, 0, 2, E21] | 5 | 210 |
| P1000 | [+, 314, E-2] | [-, 602, E21] | 3 | 1100 |
| B1999 | [314, E-2] | [-602, E21] | 2 | 2000 |
| FP15 | [FP314/-2] | [FP-602/21] | 1 | 30000 |

Table 1: **Four encodings for matrix coefficients.**

Choosing an encoding is a trade-off. Long encodings (P10, P1000) use a small vocabulary, and embed knowledge about numbers that the model can use (e.g. that numbers can be crudely compared from their signs and exponents only, that addition and multiplication can be learned by memorizing small tables). Compact encodings use a larger vocabulary (harder to learn) but result in shorter sequences that facilitate training with transformers. In P10, a $20 \times 20$ matrix is a sequence of 2002 tokens, close to the practical limit of transformers with quadratic attention. In FP15, it is only 402 tokens long.

The decision to round matrix coefficients to three significant digits is mainly motivated by the need to keep FP15 vocabulary at sizes that small transformers can learn without pre-training. Experiments about the impact of number precision can be found in appendix C.

## 2.2 Random matrix generation

In most experiments, training and test data are random matrices with coefficients uniformly distributed in $[-A, A]$ (with $A = 10$). When symmetric, these matrices are known as Wigner matrices. Their eigenvalues have a centered distribution with standard deviation $\sigma = A\sqrt{n/3}$ (see Mehta (2004) and Appendix H) that converges as $n$ grows to the semi-circle law $p(\lambda) = \sqrt{4\sigma^2 - \lambda^2}/(2\pi\sigma^2)$. If the coefficients follow a gaussian distribution, the associated eigenvectors are uniformly distributed over the unit sphere.

In section 4.4, while investigating out-of-distribution generalization, I will need to generate random symmetric matrices with specific eigenvalue distributions (i.e. classes of random matrices with non-independent coefficients). To this effect, I sample random symmetric matrices $M$ with gaussian coefficients, and compute their eigenvalue decomposition $M = PDP^T$, with $P$ an orthogonal matrix of eigenvectors (uniformly distributed over the unit sphere because the coefficients are gaussian). Replacing $D$, the diagonal matrix of eigenvalues of $M$, with a diagonal $D'$ sampled from a different distribution, and recomputing $M' = PD'P^T$, yields a symmetric matrix (since $P$ is orthogonal) with eigenvalues following the desired distribution, and eigenvectors uniformly distributed over the unit sphere.

## 3 Models and experimental settings

**Models and training.** All models use the transformer architecture from Vaswani et al. (2017): an encoder and a decoder connected by cross-attention. Models have 512 dimensions, 8 attention heads and up to 6 layers (experiments with larger models can be found in Appendix D.3). Training is supervised, minimizes the cross-entropy between model predictions and correct solutions, and uses the Adam optimiser (Kingma & Ba, 2014) with a learning rate of $10^{-4}$, a linear warm-up phase of 10,000 steps and cosine scheduling (Loshchilov & Hutter, 2016). Training data is generated on the fly in batches of 64. All models are trained on an internal cluster, using NVIDIA Volta GPU with 32GB memory. Basic operations on matrices and eigenvalues train on 1 GPU in less than a day (from a few hours for transposition and addition, to a day for multiplication and eigenvalues). Eigenvectors, SVD and inversion train on 4 GPU, and take from 3 days to a week.

**Evaluation.** At the end of every epoch (300,000 examples), a random test set (10,000 examples) is generated and model accuracy is evaluated. A predicted sequence is a correct solution to the problem $(I, O)$ ($I$ and $O$ the input and output matrices) if it can be decoded as a valid matrix $P$ and approximates the correct solution to a given tolerance $\tau$. In most problems, I check that $P$ verifies $\|P - O\| < \tau\|O\|$. When computing eigenvectors, I verify that the predicted solution $(Q, D)$ can reconstruct the input matrix, $\|QIQ^T - D\| < \tau\|D\|$. For singular value decomposition, I check that $\|UIV - S\| < \tau\|S\|$, and for matrix inversion, that $\|PI - Id\| < \tau\|Id\| = \tau$.

The $L^1$ norm, $\|A\| = \sum_{i,j} |a_{i,j}|$, for $A = (a_{i,j})$, is used in all experiments. With other norms, like $L^2$ or $L^\infty$, the error is weighted in favor of correct predictions of the largest coefficients in the solution. For eigenvalue and singular value prediction, this amounts to finding the largest values, a different and easier problem. More discussion and comparisons between norms can be found in Appendix B.

**Numerical tolerance.** All results are provided with tolerance $\tau$ between 0.5 and 5%. Since coefficients are rounded to three significant digits, 0.5% is the best we can achieve when computations are subject to rounding error. As computations become more complex, error accumulates, and larger values of $\tau$ should be considered. I consider $\tau = 0\%$ for transposition, $\tau = 1\%$ for basic matrix operations (addition and multiplication), and $\tau = 2$ or 5% for non linear operations (decomposition, inversion).

**Problem size.** All experiments are performed on dense matrices. In most cases, I focus on $5 \times 5$ matrices (or rectangular matrices with as many coefficients: e.g. $6 \times 4, 2 \times 13$), and scale to larger dimensions, from $8 \times 8$ to $15 \times 15$, and datasets of matrices with variable dimensions (e.g. $5 \times 5$ to $15 \times 15$). In this paper, the emphasis is on problems that can be solved by small transformers (up to 6 layers). I discuss scaling and larger models in Appendix D.3.

## 4   Experiments and results

This section presents experimental results for the nine problems considered. I compare encodings for different matrix sizes and tolerance levels, using the best choice of hyperparameters for each problem (i.e. the smallest architecture that can achieve high accuracy). I also show that our models are robust to noise in the training data. Learning curves and experiments with model size can be found in Appendix D, alternative architectures in Appendix E.1 (LSTM and GRU) and E.2 (universal transformers), and additional tasks (re-training, joint training) in Appendix F.

### 4.1   Transposition

Learning to transpose a matrix amounts to learning a permutation of its elements. For a square matrix, all cycles in the permutation have length 1 or 2. Longer cycles may appear in rectangular matrices. This task involves no arithmetic operations: tokens in the input sequence are merely copied to different positions in the output. This paper investigates two cases. In the fixed-dimension case, all matrices in the dataset have the same dimensions and only one permutation must be learned. In the variable-dimension case, the dataset includes matrices of different formats, and several permutations must be learned (one per matrix format). In these experiments, models have one layer, 256 dimensions and 8 attention heads, and use the four encodings.

After training, all models achieve 99% exact accuracy (0% tolerance) for fixed-size matrices with dimensions up to $30 \times 30$. This holds for all encodings and input and output sequence lengths up to 2000 tokens. The variable-size case proves more difficult, because the model must learn many different permutations. Still, the model achieve 99% accuracy on matrices with 5 to 15 dimensions, and 96% for matrices with 5 to 20 dimensions. Table 2 summarizes the results.

| | Fixed dimensions | | | | | | | Variable dimensions | | | |
| | | | | | | | | Square | | Rectangular | |
| | 5x5 | 10x10 | 20x20 | 30x30 | 5x6 | 7x8 | 9x11 | 5-15 | 5-20 | 5-15 | 5-20 |
|---|---|---|---|---|---|---|---|---|---|---|---|
| P10 | 100 | 100 | 100 | - | 100 | 100 | 100 | 100 | - | 97.0 | - |
| P1000 | 100 | 100 | 99.9 | - | 100 | 100 | 100 | 99.9 | - | 98.4 | - |
| B1999 | 100 | 100 | 99.9 | 100 | 100 | 100 | 100 | 100 | 96.6 | 99.6 | 91.4 |
| FP15 | 99.8 | 99.5 | 99.4 | 99.8 | 99.8 | 99.5 | 99.3 | 99.8 | 99.6 | 99.4 | 96.1 |

Table 2: **Exact prediction of matrix transposition for different matrix dimensions.** Transformers with one layer, 256 dimensions and 8 attention heads.

## 4.2 Addition

To add two $m \times n$ matrices, the model must learn the correspondence between input and output positions and the algorithm for adding two numbers in scientific notation. Then, it must apply the algorithm to $mn$ pairs of coefficients. In these experiments, models have one or two layers, 8 attention heads and 512 dimensions.

All models achieve 99% accuracy at 1% tolerance (98% at 0.5%) on sums of fixed-size matrices with dimensions up to $10 \times 10$, for all four encodings. B1999 models achieve 99.5% accuracy at 0.5% tolerance for $15 \times 15$ matrices and 87.9% accuracy at 1% tolerance on $20 \times 20$ matrices. As dimensions increase, models using long encodings (P1000 and P10) become more difficult to train as their input sequences grow longer. For instance, adding two $15 \times 15$ matrices involves 450 coefficients, an input of 1352 tokens in P1000 and 2252 in P10.

On variable-size matrices, models achieve 99.5% accuracy at 1% tolerance for dimensions up to 10, with 2-layer transformers using the B1999 encoding. Their accuracy drops to 48 and 37% for square and rectangular matrices with 5 to 15 dimensions. This can be mitigated by increasing the depth of the decoder: models with one layer in the encoder and 6 in the decoder achieve 77 and 87% accuracy. Table 3 summarizes these results.

| | Fixed dimensions | | | | | | Variable dimensions | | | | | |
| | | | | | | | Square | | | Rectangular | | |
| Size | 5x5 | 6x4 | 3x8 | 10x10 | 15x15 | 20x20 | 5-10 | 5-15 | 5-15 | 5-10 | 5-15 | 5-15 |
| Layers | 2/2 | 2/2 | 2/2 | 2/2 | 2/2 | 1/1 | 2/2 | 1/1 | 1/6 | 2/2 | 2/2 | 1/6 |
|---|---|---|---|---|---|---|---|---|---|---|---|---|
| 5% | 100 | 99.9 | 99.9 | 100 | 100 | 98.8 | 100 | 63.1 | 99.3 | 100 | 72.4 | 99.4 |
| 2% | 100 | 99.5 | 99.8 | 100 | 100 | 98.4 | 99.8 | 53.3 | 88.1 | 99.8 | 50.8 | 94.9 |
| 1% | 100 | 99.3 | 99.7 | 100 | 99.9 | 87.9 | 99.5 | 47.9 | 77.2 | 99.6 | 36.9 | 86.8 |
| 0.5% | 100 | 98.1 | 98.9 | 100 | 99.5 | 48.8 | 98.9 | 42.6 | 72.7 | 99.1 | 29.7 | 80.1 |

Table 3: **Accuracies of matrix sums, for different tolerances.** B1999 encoding, 512 dimension and 8 attention heads.

## 4.3 Multiplication

Multiplication of a matrix $M$ of dimension $m \times n$ by a vector $V \in \mathbb{R}^n$ amounts to computing $m$ dot products between $V$ and the lines of $M$. Each calculation features $n$ multiplications and $n-1$ additions, and involves one row in the matrix and all coefficients in the vector. The model must now learn two operations: add and multiply. Experiments with models with 1 and 2 layers show that high accuracy can only be achieved with the P10 or P1000 encoding, with P1000 performing better on average. The number of layers, on the other hand, makes little difference.

| | P10 | P1000 | | P1000 | | | | Variable 5-10 (P1000) | |
| | 5x5 | 5x5 | 10x10 | 14x2 | 9x3 | 4x6 | 2x10 | Square | Rectangular |
| Tolerance | 2/2 layers | 2/2 | 2/2 | 1/1 | 1/1 | 2/2 | 2/2 | 4/4 | 2/2 |
|---|---|---|---|---|---|---|---|---|---|
| 5% | 100 | 100 | 100 | 99.3 | 99.9 | 100 | 100 | 72.4 | 41.7 |
| 2% | 99.9 | 100 | 100 | 99.0 | 99.7 | 100 | 99.8 | 68.4 | 35.0 |
| 1% | 98.5 | 99.9 | 99.9 | 98.7 | 99.5 | 99.9 | 99.2 | 60.1 | 20.1 |
| 0.5% | 81.6 | 99.5 | 98.4 | 98.1 | 99.0 | 98.6 | 94.5 | 30.8 | 4.4 |

Table 4: **Accuracies of matrix-vector products, for different tolerances.** All model have 512 dimensions and 8 heads.

On this task, models achieve 99.9% accuracy at 1% tolerance for $5 \times 5$ and $10 \times 10$ square matrices, and 99% for rectangular matrices with about 30 coefficients. The variable-size case proves much harder. Models achieve non-trivial results: 60% accuracy with 1% tolerance for square matrices, but larger models are needed for high accuracy. Table 4 summarizes the results.

Multiplication of matrices $M$ and $P$ is a scaled-up version of matrix-vector multiplication, now performed for every column in matrix $P$. As above, high accuracy is only achieved with the P10 and P1000 encoding.

| | Square matrices | | Rectangular matrices | | | | | | | |
|---|---|---|---|---|---|---|---|---|---|---|
| | 5x5 | 5x5 | 2x13 | 2x12 | 3x8 | 4x6 | 6x4 | 8x3 | 12x2 | 13x2 |
| Tolerance | P10 2/2 layers | 1/4 | 4/4 | 4/4 | 2/6 | 1/4 | 1/6 | 1/6 | 1/6 | 1/4 |
| 5% | 100 | 100 | 100 | 100 | 100 | 100 | 100 | 100 | 100 | 99.9 |
| 2% | 100 | 100 | 100 | 100 | 100 | 100 | 100 | 100 | 99.7 | 99.8 |
| 1% | 99.8 | 100 | 99.9 | 100 | 100 | 99.9 | 100 | 99.9 | 99.3 | 99.8 |
| 0.5% | 64.5 | 99.9 | 97.1 | 98.5 | 99.6 | 99.7 | 99.5 | 99.5 | 99.0 | 99.8 |

Table 5: **Accuracy of matrix multiplication, for different tolerances.** Fixed-size matrices with 24-26 coefficients. All encodings are P1000 unless specified. Models have 512 dimensions and 8 attention heads.

Models achieve 99% accuracy at 1% tolerance for $5 \times 5$ square matrices and rectangular matrices of comparable dimensions (see Table 5). Performance is the same as matrix-vector multiplication, a simpler task. However, matrix multiplication needs deeper models (especially decoders), and more training time.

### 4.4 Eigenvalues

Compared to basic operations on matrices, computing the eigenvalues of symmetric matrices is a much harder problem, non-linear and typically solved by iterative algorithms. Deeper models, with 4 or 6 layers, are used in this task. They achieve 100% accuracy at 5% tolerance, and 99% at 2%, for $5 \times 5$ and $8 \times 8$ matrices. High accuracy is achieved with all four encodings, but P1000 proves more efficient with $8 \times 8$ matrices.

On fixed-size datasets, scaling to larger problems proves difficult. It takes 360 million examples for our best models to reach 25% accuracy on $10 \times 10$ matrices. As a comparison, 40 million examples are required to train $5 \times 5$ models to 99% accuracy, and 60 million for $8 \times 8$ models. This limitation can be overcome by training on variable-size datasets, achieving 100% accuracy at 5% tolerance, and $100, 100$ and 76% at 2%, for sets of 5-10, 5-15 and 5-20 matrices. Table 6 summarizes the results.

| | Fixed dimensions | | | | | | | Variable dimensions | | |
|---|---|---|---|---|---|---|---|---|---|---|
| | 5x5 | 5x5 | 5x5 | 5x5 | 8x8 | 8x8 | 10x10 | 5-10 | 5-15 | 5-20 |
| Encoding | P10 | P1000 | B1999 | FP15 | P1000 | FP15 | FP15 | FP15 | FP15 | FP15 |
| Layers | 6/6 | 4/1 | 6/6 | 6/1 | 6/1 | 1/6 | 1/6 | 4/4 | 6/6 | 4/4 |
| 5% | 100 | 100 | 100 | 100 | 100 | 100 | 25.3 | 100 | 100 | 100 |
| 2% | 100 | 99.9 | 100 | 100 | 99.2 | 97.7 | 0.4 | 99.8 | 100 | 75.5 |
| 1% | 99.8 | 98.5 | 98.6 | 99.7 | 84.7 | 77.9 | 0 | 87.5 | 94.3 | 45.3 |
| 0.5% | 93.7 | 88.5 | 73.0 | 91.8 | 31.1 | 23.9 | 0 | 37.2 | 40.6 | 22.5 |

Table 6: **Accuracy of eigenvalues for different tolerances and dimensions.** All models have 512 dimensions and 8 attention heads, except the 10x10 model, which has 510 and 12.

**Larger models.** In the fixed-dimension case, 6-layer models are limited to $8 \times 8$ matrices. Experiments with deeper models show that they can solve larger problems. For instance, 12-layer transformers can compute the eigenvalues of $12 \times 12$ matrices. Large models also need less examples to train to high accuracy. Table 7 summarizes these results. Detailed results are in Appendix D.3

| | Accuracy | | | Sample size (millions) | | |
|---|---|---|---|---|---|---|
| Layers | 1/8 | 1/12 | 1/24 | 1/8 | 1/12 | 1/24 |
| $8 \times 8$ matrices | 100 | 100 | 100 | 28.8 | 11.4 | 11.1 |
| $10 \times 10$ matrices | 100 | 100 | 100 | 85.2 | 36.6 | 32.1 |
| $12 \times 12$ matrices | 3 | 97 | 100 | - | - | 99.3 |

Table 7: **Eigenvalues, larger models.** Accuracy to 5% tolerance, and sample size to reach 99% accuracy.

### 4.5 Eigenvectors

In this task, the model predicts both the eigenvalues and an associated orthogonal matrix of eigenvectors. Models using the P10 and P1000 encoding achieve 97 and 94% accuracy at 5% tolerance for $5 \times 5$ matrices. P1000 models also reach 82% accuracy on $6 \times 6$ matrices. Whereas FP15 models only reach 52% accuracy, an asymmetric model, coupling a 6-layer FP15 encoder and a 1-layer P1000 decoder, achieves 94% accuracy at 5% and 87 at 2%, the best result on this task. Table 8 summarizes these results.

| | 5x5 | | | | 6x6 |
|---|---|---|---|---|---|
| | P10 4/4 layers | P1000 6/6 | FP15 1/6 | FP15/P1000 6/1 | P1000 6/1 |
| 5% | 97.0 | 94.0 | 51.6 | 93.5 | 81.5 |
| 2% | 83.4 | 77.9 | 12.6 | 87.4 | 67.2 |
| 1% | 31.2 | 41.5 | 0.6 | 67.5 | 11.0 |
| 0.5% | 0.6 | 2.9 | 0 | 11.8 | 0.1 |

Table 8: **Accuracies of eigenvectors, for different tolerances and depths** (512 dimensions, 8 heads).

**Analysis of failure cases.** On this task, models achieve significantly less than 100% accuracy. This makes it possible to investigate failure cases. For this analysis, I use the trained FP15/P1000 6/1 layer model from table 8 (93% accuracy), generate a new test sample of 10 000 problems, predict solutions, and evaluate performance on various metrics. On this new test set, the model achieves 91.1% accuracy with 5% tolerance, and 82.1, 45.2 and 1.4% at 2, 1 and 0.5 tolerance.

First, one notes that almost all model predictions (9999 out of 10000) are well-formed matrices: the model produces no meaningless output, such as incorrect encoding of numbers, or matrices with the wrong number of elements. This is consistently observed in all experiments: output syntax is learned to near-perfection at the beginning of training. Also, accuracy increases with tolerance: 95.6% accuracy at 25% tolerance. This suggests that, even when they fail to predict the correct solution, models do not hallucinate irrelevant predictions (as was reported on natural language tasks). Instead, they predict syntactically correct solutions that often turn out to be "rough approximations" of the correct answer.

When computing eigenvectors, the model predicts two matrices, a diagonal matrix of eigenvalues $D$ and an orthogonal matrix of eigenvectors $H$. Accuracy is measured as the $L^1$ distance between $H^T IH$ ($I$ the input matrix) and $D$, i.e. how well $H$ diagonalizes the input into $D$. A different metric, the distance between $HDH^T$ and $I$, could be used instead, which is associated to a related, but weaker, problem: finding approximations to $I$ of the form $HDH^T$ ($D$ diagonal). With this metric, the model achieves slightly higher accuracy: 95.7% at 5% tolerance (92.9, 88.4 and 73.0 at 2, 1 and 0.5%). This confirms our previous observation that, in many failure cases, the model predicts solutions that are "somehow relevant" to eigen decomposition (here, solutions to the weak problem).

We also know from theory that $D$ should contain the eigenvalues of the input matrices, and that $H$ should be orthogonal (all lines and columns orthogonal, with unit norm). Translating these properties into metrics can help us understand failure cases, and the relevance of incorrect model predictions.

First, eigenvalues are always correctly predicted: the corresponding accuracy is 100% at 5% tolerance, and 99.4% at 0.5%. The (easier) sub-task of eigenvalue prediction has been learned by the model. Second, all the norms of the columns of $H$ are within 5% of 1 in 99.9% of the test examples (within 1% in 99.2%). All predicted eigenvectors have unit norm. This indicates that failed model predictions actually succeed in computing the eigenvalues, and a set of unit vectors. In other words, those two properties of eigen decomposition have been learned by the model, in the sense that they are respected even in incorrect predictions.

To measure the orthogonality of the eigenvectors, I compute the dot products of successive eigenvectors, which should be all be 0. On the test set, all dot products are within $-0.05$ and $0.05$ in 93.6 of the cases (and within $-0.01$ and $0.01$ in 85.1). This suggests that when the model fails, it proves the correct eigenvalues, and unit eigenvectors, but fails to make the "eigenvectors" strictly orthogonal. This observation suggests a criterion for

predicting model accuracy: we can test the orthogonality of predicted $H$ by measuring its condition number (the ratio of its largest and smallest singular values), which should be one if $H$ is orthogonal, and larger if it is not. In fact, in 98% of correct predictions, the predicted $H$ has a condition number smaller than 1.035. For 98% of failures, the condition number of $H$ is larger than 1.04.

To summarize, when trained on eigen decomposition, the model learns the easier sub-task of predicting eigenvalues, with 100% accuracy. It also learns to preserve theoretical properties of the result, like the unit norm of eigenvectors, and their (approximate) orthogonality. All failures concentrate on one specific sub-task: orthogonalizing the eigenvectors. This allows us to derive an accurate predictor of model failure: the condition number of the predicted matrix $H$.

### 4.6 Inversion

Computing the inverses of $5\times5$ matrices proves the hardest task so far. Models using the P10 and P1000 encodings, with 6-layer encoders and 1-layer decoders and 8 attention heads, achieve 74 and 80% accuracy at 5% tolerance. Adding more heads in the encoder bring no gain in accuracy, but makes training faster: 8-head models need 250 millions examples to train to 75% accuracy, 10 and 12-head models only 120. As in the previous task, asymmetric models achieve the best results. 90% accuracy at 5% tolerance can be reached with a 6-layer FP15 encoder with 12 attention heads, and a 1-layer P1000 decoder with 8 heads.

| | P10 | P1000 | | | FP15/P1000 | |
| Tolerance | 8/8 heads | 8/8 heads | 10/8 heads | 12/8 heads | 10/4 heads | 12/8 heads |
| --- | --- | --- | --- | --- | --- | --- |
| 5% | 73.6 | 80.4 | 78.8 | 76.9 | 88.5 | 90.0 |
| 2% | 46.9 | 61.0 | 61.7 | 52.5 | 78.4 | 81.8 |
| 1% | 15.0 | 30.1 | 34.2 | 16.2 | 55.5 | 60.0 |
| 0.5% | 0.2 | 3.1 | 5.9 | 0.1 | 20.9 | 24.7 |

Table 9: **5x5 matrix inversion.** All models have 512 dimension and 6/1 layers, except P1000 10 heads, which has 6/6.

**Analysis of failure cases.**  I proceed as in section 4.5, and use the 6/1 layer, 12/8 heads, FP15/P1000 model from table 9 (90% accuracy). Model accuracy on the new test set is 89.6% at 5% tolerance, and 81.7, 59.1 and 23.7 at $2, 1$ and 0.5% tolerance. As previously, all 10000 model predictions are well-formed matrices, and accuracy increases with tolerance: 96.6% at 25%. Again, even when the model fails, it provides a "relevant" bad approximation of the solution, instead of hallucinating an unrelated guess.

For this task, the accuracy metric is the distance between $PI$ ($P$ the predicted matrix, $I$ the input) and identity (in $L^1$ norm). This measures that the inverse has indeed been found. However, since the inverse is unique, we might as well use the distance between $P$ and $I^{-1}$ (i.e. the distance the model is minimizing during training). On this alternative metric, the model achieves 98.2% accuracy at 5% tolerance, and 96.0, 92.3 and 84.5% at $2, 1$ and 0.5 tolerance. In this metric, all failure cases are bad approximations: the model achieves 99.5% accuracy at 25% tolerance.

This suggests that most model failures happen because the approximation of the $I^{-1}$ predicted by the model is not a "good inverse" of $I$, in the sense that $PI$ is not close to identity. Theory tells us this happens when the condition number of the input matrix (the ratio of largest to smallest singular values) is large. Indeed, 98% of correct predictions correspond to matrices with condition number below 51.5. On the other hand, 98% of failures are matrices with condition numbers larger than 51.5. The condition number of the input matrix proves to be a very accurate predictor of model success.

These results provide a complete explanation of model failures for this task. They indicate that failures are not due to the architecture or learning technique, but to the mathematical limitations of the computation of matrix inverses, which apply to every numerical algorithm. They also indicate that failures are concentrated on a small class of problems, and can be predicted in advance (without running the model, in this case).

They also suggest two directions for improvement. First, we could oversample ill-conditioned matrices in the training set, in a manner of curriculum learning. Second, since ill-conditioning amplifies the effect of rounding and approximate computations, training with increased precision should improve accuracy.

## 4.7   Singular value decomposition (SVD)

For symmetric matrices, singular value and eigenvalue decompositions are related: the singular values of a symmetric matrix are the square roots of the absolute values of its eigenvalues, and the vectors are the same. Yet, this task proves more difficult than computing the eigenvectors. Models achieve 100 accuracy at 5% tolerance, and 86.7% at 1% when predicting the singular values of $4 \times 4$ symmetric matrices. For the full decomposition, models achieve 98.9 and 75.3% accuracy. However, the SVD of $5 \times 5$ matrices could not be predicted using transformers with up to 6 layers, and using the P10 or P1000 encoding. Table 10 summarizes these results, on models with 512 dimensions and 8 attention heads.

| | Singular values | | Singular vectors | |
|---|---|---|---|---|
| | P10 2/2 layers | P1000 4/4 layers | P10 1/6 layers | P1000 6/6 layers |
| 5% | 100 | 100 | 71.5 | 98.9 |
| 2% | 98.5 | 99.8 | 15.6 | 95.7 |
| 1% | 84.5 | 86.7 | 0.4 | 75.3 |
| 0.5% | 41.1 | 39.8 | 0 | 6.3 |

Table 10: **Accuracies of SVD for 4x4 matrices.**

## 4.8   Experiments with noisy data

Because experimental data is often noisy, robustness to noise is a key feature of efficient models. In this section, I investigate model behavior in the presence of random error when computing the sum and eigenvalues of $5 \times 5$ matrices. A random gaussian error is added to all coefficients of the input matrices in the train and test sets, for three levels of noise: standard deviation equal to $1, 2$ and 5% of the standard deviation of the random matrix coefficients ($\sigma = 5.77$ for uniform coefficients in $[-10, 10]$). For a linear operation like addition, one expects the model to predict correct results so long tolerance $\tau$ is larger than error. For non-linear computations like eigenvalues, expected outcomes are unclear, as errors may be amplified by non-linearities or reduced by concentration laws.

| Encoding | Addition B1999 | | Eigenvalues FP15 | | P1000 | |
|---|---|---|---|---|---|---|
| Dimension | 256 | 512 | 512 | 1024 | 512 | 1024 |
| 5% tolerance | | | | | | |
| $0.01\sigma$ error | 100 | 100 | 6.1 | 100 | 100 | 100 |
| $0.02\sigma$ | 100 | 100 | 100 | 100 | 100 | 100 |
| $0.05\sigma$ | 41.5 | 41.2 | 99.1 | 99.3 | 99.3 | 99.0 |
| 2% tolerance | | | | | | |
| $0.01\sigma$ error | 99.8 | 99.9 | 0.7 | 99.8 | 99.3 | 99.6 |
| $0.02\sigma$ | 43.7 | 44.2 | 97.0 | 97.1 | 97.3 | 97.9 |
| $0.05\sigma$ | 0 | 0 | 37.9 | 38.4 | 40.1 | 37.3 |
| 1% tolerance | | | | | | |
| $0.01\sigma$ error | 39.8 | 41.7 | 0.1 | 82.1 | 79.7 | 83.8 |
| $0.02\sigma$ | 0.1 | 0.1 | 47.8 | 51.3 | 46.2 | 47.5 |
| $0.05\sigma$ | 0 | 0 | 3.8 | 4.2 | 4.1 | 3.8 |

Table 11: **Accuracy with noisy data, for different error levels and tolerances ($5 \times 5$ matrices).**

**Addition.** Training on noisy data causes no loss in accuracy in the models, so long the ratio between the standard deviation of noise and that of the coefficients is lower than tolerance. Within 5% tolerance, models

trained with $0.01\sigma$ and $0.02\sigma$ noise reach 100% accuracy, as do models trained with trained with $0.01\sigma$ noise at 2% tolerance. Accuracy drops to about 40% when error levels are approximately equal to tolerance, and to zero once error exceeds tolerance. Model size and encoding have no impact on robustness (see Table 11, 2-layer, 8-head models and Table 27 in Appendix F.3).

**Eigenvalues.** Models trained with the P1000 encoding prove more robust to noise when computing eigenvalues than when calculating sums. For instance, they achieve 99% accuracy at 5% tolerance with noise equal to $0.05\sigma$, vs only 41% for addition. As before, model size has no impact on robustness. However, FP15 models prove more difficult to train on noisy data than P1000 (see Table 11 and Table 28 in Appendix F.3 for additional results, models have 4 layers and 8 heads).

## 5  Out-of-domain generalization

So far, model accuracy was measured on test sets of matrices generated with the same procedure as the training set. In this section, I investigate accuracies on test sets with different distributions. I focus on one task: predicting the eigenvalues of symmetric matrices (with tolerance 2%).

**Wigner matrices.** All models are trained on datasets of random symmetric real matrices, with independent and identically distributed (iid) coefficients sampled from a uniform distribution over $[-A, A]$. These are known as Wigner matrices (see 2.2), and constitute a very common class of random matrices. Yet, matrices with different eigenvalue distributions (and non iid coefficients) appear in important problems. For instance, statistical covariance matrices have all their eigenvalues positive, and the adjacency matrices of scale-free and other non-Erdos-Renyi graphs have centered but non semi-circle distributions of eigenvalues (Preciado & Rahimian, 2017). We now investigate how models trained on Wigner matrices perform on test sets of matrices with different distributions.

**Testing on different distributions.** Matrix coefficients in the training set are sampled from $\mathcal{U}[-10, 10]$, with standard deviation $\sigma_{tr} = 5.77$. First, I consider test sets of Wigner matrices with different standard deviation $\sigma_{tst}$. Models achieve high accuracy (96% at 2% tolerance) so long $0.6\sigma_{tr} < \sigma_{tst} < \sigma_{tr}$. Out of this range, model accuracy drops: to 54% for $0.4\sigma_{tr}$, to 26% for $1.1\sigma_{tr}$, to 2% for $1.3\sigma_{tr}$ and to 0% for $0.2\sigma_{tr}$. Then, the model is tested on sets of matrices with different eigenvalue distributions: positive, uniform, Gaussian and Laplace (generated as per section 2.2), with standard deviation $\sigma_{tr}$ and $0.6\sigma_{tr}$. With $\sigma_{tst} = \sigma_{tr}$, the model achieves 26% accuracy for Laplace, 25 for Gaussian, 19 for uniform, and 0 for positive. With $\sigma_{tst} = 0.6\sigma_{tr}$, model accuracy is slightly higher, $28, 44, 60$ and 0% respectively, but remains low overall. Matrices with positive eigenvalues cannot be predicted at all. These results are summarized in line 1 of Table 12. These results confirm previous observations (Welleck et al., 2021): transformers only generalize to a narrow neighborhood around their training distribution.

**Training on different distributions.** A common approach to improving out-of-distribution accuracy is to make the training set more diverse. Models trained from a mixture of Wigner matrices with different standard deviation ($A \in [1, 100]$, line 2 of Table 12) generalize to Wigner matrices of all standard deviation (which are no longer out-of-distribution), and achieve better performances on the uniform, Gaussian and Laplace test set. But they do not generalize to positive matrices. A model trained on a mixture of Wigner and positive eigenvalues (line 3 of Table 12) can predict positive eigenvalues (now in-domain), but its performance degrades on all other test sets.

Training on mixtures of Wigner and Gaussian eigenvalues, or Wigner and Laplace eigenvalues (lines 4 and 5 of Table 12), achieves high accuracies over all test sets, including the out-of-distribution sets: uniform and positive eigenvalues, and Wigner with low or high standard deviations.

Finally, models trained on matrices with Laplace eigenvalues only, or a mixture of uniform, Gaussian and Laplace eigenvalues (all non-Wigner matrices) achieve 95% accuracy over all test sets (lines 6 and 7 of Table 12). These results confirm that out-of-distribution generalization is possible, if attention is paid to the training data distribution. They also suggest that Wigner matrices, the default model for random matrices, is not the best choice for training transformers: models trained on Wigner matrices do not generalize out of distribution, whereas models trained on non-Wigner matrices, with non-iid coefficients, do generalize to Wigner matrices.

| Train set distribution | Test set eigenvalue distribution | | | | | | | | | | |
| --- | --- | --- | --- | --- | --- | --- | --- | --- | --- | --- | --- |
| | Wigner | | | Positive | | Uniform | | Gaussian | | Laplace | |
| $\sigma_{tst}/\sigma_{tr}$ | 0.3 | 1.0 | 1.2 | 0.6 | 1 | 0.6 | 1 | 0.6 | 1 | 0.6 | 1 |
| Wigner, A=10 (baseline) | 12 | 100 | 7 | 0 | 0 | 60 | 19 | 44 | 25 | 28 | 26 |
| Wigner, $A \in [1, 100]$ | 99 | 98 | 97 | 0 | 0 | 68 | 60 | 65 | 59 | 57 | 53 |
| Wigner - Positive | 1 | 99 | 14 | 88 | 99 | 45 | 23 | 31 | 23 | 17 | 20 |
| Wigner - Gaussian | 88 | 100 | 100 | 99 | 99 | 96 | 98 | 93 | 97 | 84 | 90 |
| Wigner - Laplace | 98 | 100 | 100 | 100 | 100 | 100 | 100 | 99 | 100 | 96 | 99 |
| Laplace | 95 | 99 | 99 | 100 | 100 | 98 | 98 | 97 | 98 | 94 | 96 |
| Gaussian-Uniform-Laplace | 99 | 100 | 100 | 100 | 100 | 100 | 100 | 99 | 100 | 97 | 99 |

Table 12: **Out-of-distribution eigenvalue accuracy (tolerance** $2\%$**) for different training distributions**. All models have 512 dimensions and 8 attention heads, and use the P1000 encoding.

# 6 Related work

**Neural networks for linear algebra.** Neural networks that can compute eigenvalues and eigenvectors have been proposed since the early 1990s (Samardzija & Waterland, 1991; Cichocki & Unbehauen, 1992; Oja, 1992; Yi et al., 2004), and are still an active field of research (Tang & Li, 2010; Finol et al., 2019). They leverage the Universal Approximation Theorem (Cybenko, 1989; Hornik, 1991), which states that, under weak conditions on their activation functions, neural networks can approximate any continuous mapping – in this case, the mapping between a matrix and its eigenvalues or vectors. In these works, the network represents a differential equations involving matrix coefficients, which features the eigenvalues in its solution (Brockett, 1991). The matrix to decompose is encoded in the input, and prediction errors are back-propagated until a solution to the differential equation is found, from which eigenvalues can be recovered. Note that these models compute their solutions during training, and must be retrained every time a new matrix is to be processed. Similar techniques have been proposed for other problems of linear algebra (Wang, 1993a;b; Zhang et al., 2008).

**Arithmetic with neural networks.** Neural networks for binary addition and multiplication have been proposed since the 1990s (Siu & Roychowdhury, 1992). Since 2015, recurrent architectures have been used, from LSTM (Kalchbrenner et al., 2015) to RNN (Zaremba et al., 2015), Neural Turing Machines (Castellini, 2019) and Neural GPU (Kaiser & Sutskever, 2015). All authors note that sequential models struggle to generalize out of their training distribution (i.e. to larger numbers), and that their architectures only perform satisfactorily on binary numbers. Neural Arithmetic Logic Units (NALU (Trask et al., 2018) were introduced as a solution to the generalization problem. They can perform exact additions, substractions, multiplications and divisions by constraining the weights of a linear network to remain close to 0, 1 or -1. NALU (and Neural GPU) can extrapolate to numbers far larger than those they were trained on, and could serve as building blocks for larger models. The use of language models for arithmetic and problem solving has been studied by Saxton et al. (2019). Palamas (2017) experiments with modular arithmetic.Nogueira et al. (2021) investigates the limitations of transformers.

**Transformers for mathematics.** Early applications of transformers to mathematics focused on symbolic computation. Lample & Charton (2019) used transformers to compute symbolic integrals and solve differential equations. Davis (2019) and Welleck et al. (2021) discuss the limits of their approach, especially with respect to out-of-distribution generalization. Transformers have also been applied to theorem proving (Polu & Sutskever, 2020; Han et al., 2021), temporal logic (Hahn et al., 2021), and have been proposed as a replacement for the genetic algorithms used in symbolic regression (Biggio et al., 2021; d'Ascoli et al., 2022). In more numerical applications, Charton et al. (2020) use them to predict the numerical properties of differential systems, and Dersy et al. (2022) to simplify formulas involving polylogarithms.

With the advent of large language models (Bommasani et al., 2021), a new line of research focuses on informal mathematics: solving problems of mathematics written in natural language, as a language task (Griffith & Kalita, 2021; Meng & Rumshisky, 2019; Cobbe et al., 2021). Lewkowycz et al. (2022) show that a very large

(540 billion parameters) pre-trained transformer can be retrained on a large math corpus to solve grade and high school problems of mathematics. Welleck et al. (2022) apply similar techniques to theorem proving.

**Other architectures for mathematics.** Graph Neural Networks (Scarselli et al., 2009) have been widely used in scientific applications of AI, because of their capacity to integrate problem or domain-specific inductive biases into the network structure. They have been applied to a wide range of mathematical problems, from dynamical systems (Iakovlev et al., 2020) to combinatorial optimization (Cappart et al., 2021) and knot theory (Davies et al., 2021). Vinyals et al. (2015) proposed pointer networks to solve combinatorial problems. Blalock & Guttag (2021) use machine learning techniques to improve existing algorithms for matrix multiplication, in the specific case where one fixed matrix should be multiplied by many others.

## 7 Discussion

**Encodings and architecture.** Our best results are achieved using the P1000 and FP15 encodings. For most problems, P10 is dominated by the more economical P1000, and B1999 never finds its use, between the more compact FP15 and the more efficient P1000. P1000 emerges as a good choice for problems of moderate size, and FP15 when sequences grow long. For the hardest problems, eigenvectors and inversion, asymmetric encodings, FP15 in the encoder and P1000 in the decoder, achieve the best results. I believe that the longer and meaningful P1000 output representation provide better error feedback to the model, facilitating learning, while the FP15 encoding provides a compact representation of the input, which is easier to train.

Experiments also showcase the efficacy of asymmetric architectures, with one layer in either the encoder or decoder. Whether the encoder or the decoder should be shallow is unclear: on the eigenvalue and eigenvector tasks, the 6/1 and 1/6 architecture seem equally efficient. Finally, increasing the number of attention heads seems to help. Most transformers architectures (from Vaswani to BERT) maintain a dimension/head ratio of 64, increased to 96 or more in very large models like GPT-3. For eigenvalue and inversion, using 10 or 12 heads with dimension 512, i.e. a dimension/head ratio between 40 and 50, improves model accuracy.

**Model limitations, scaling to large dimensions.** Most experiments feature dense matrices with 5 to 10 dimensions. Experiments with eigenvalues suggest that larger problems can be solved by training from samples of matrices of variable size, or by using larger models. However, scaling to larger dense matrices will be limited by the length of the sequences a transformer can handle. For quadratic attention models (i.e. most current transformer architectures), sequence length can hardly exceed a few thousand tokens, and the methods proposed in this paper could probably not scale beyond $50 \times 50$ matrices. Experimenting with transformers with linear or log-linear attention (Zaheer et al., 2021; Wang et al., 2020a; Vyas et al., 2020; Child et al., 2019) is a natural extension of this work. Problems of larger dimension usually feature sparse matrices, and therefore are out of the scope of this work. Extension to sparse matrices constitutes a future research direction.

**Out-of-distribution experiments.** These are our most significant results. They prove that transformers trained on random data **can** generalize to a wide range of test distributions, provided their training data distribution is chosen with care. Selecting a training distribution can be counter-intuitive. In our experiments, Wigner matrices are the "obvious" random model, but "special" matrices (with non-iid coefficients and Laplace eigenvalues) produce models that better generalize, notably on Wigner matrices. This matches the intuitive idea that we learn more from edge cases than averages.

**Result verification.** One common criticism of deep learning models is that they provide no guarantee on the correctness of their output. This limitation does not apply here, as the model achieves 100% accuracy on basic matrix operations and eigenvalue calculations, and our analysis of failure cases propose a mitigation for the harder problems of eigenvectors and matrix inversion.

**Do the models memoize?** Transformers are often accused of using the large capacity of their feed-forward networks to memorize training examples, and interpolating between them at inference. Three observations lead me to believe that this is not the case here. First, in section 4.4, the eigenvalues of matrices with more than 9 dimensions cannot be learned from a training set where all matrices have the same size. However, training on a mixture of matrices from $5 \times 5$ to $20 \times 20$ allows all dimensions to be learned. If memoization happened, a training set with just one dimension would be easier to train than a mixture. Second, in

appendix F.1, retraining a model on matrices of a different dimension takes significantly less examples than training it from scratch. In a memoization setting, there would be little benefit to retraining. Finally, the results on out-of-domain generalization seem to rule out interpolation. A model trained on matrices with Laplace distributed eigenvalues (which can be positive or negative) will generalize to positive definite matrices, a different ensemble, with very little overlap (almost no Laplace matrix is positive).

**Comparison with numerical packages.** Given the practical importance of linear algebra, optimized numerical libraries exist for most programming languages and environment. Since my models run in Python, I compare them with Numpy.

Calculating the eigenvalues of a $5 \times 5$ matrix takes 0.5 millisecond on a trained 4/1 layer transformer running pyTorch on a single GPU machine. Matrix inversion takes 1 ms with a 6/1 layer transformer, running pyTorch on a single GPU machine. On the same machine, the optimized algorithms in Numpy (linalg.eigval, and linalg.inv) are faster : 0.07 millisecond for eigenvalues and 0.04 ms for inversion. However, the current code was not designed for speed. An optimized version of the models might achieve inference speeds comparable to those of numerical packages. Note, however, that the memory footprint of a transformer would be considerably larger).

For these two tasks, the algorithms implemented in Numpy and other packages have asymptotic complexity $O(n^3)$ ($O(n^{2.37})$ for the best known bounds) for $n \times n$ matrices. The attention mechanism of the transformers used in this paper is quadratic in the length of the sequence ($O(n^2)$), which makes it $O(n^4)$. Linear attention models could reduce complexity to $O(n^2)$, lower than known algorithms, but the memory requirement of transformers would offset this advantage for large $n$. As stated in the introduction, there is no clear advantage to replace existing algorithms with transformers.

## 8    Conclusion.

I have shown that transformers can learn to perform numerical computations from examples only. I also proved that they can generalize out of domain when their training distribution is carefully selected. This suggests that applications of transformers to mathematics are not limited to symbolic computation, and can cover a broader range of scientific problems. I believe that these results pave the way for wider use of transformers in science.

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

## A    Number encodings

Let $x$ be a non-zero real number, it can be represented uniquely as $x = s.m.10^e$, with $s \in \{-1, 1\}$, $m \in [100, 1000[$ and $e \in \mathbb{Z}$. Rounding $m$ to the nearest integer $n$ (and potentially adjusting for round-up to 1000), we get the base ten, floating-point representation of $x$, with three significant digits:

$$x \approx s.m.10^e, (s, m, e) \in \{-1, 1\} \times \{100, \dots, 999\} \times \mathbb{Z}$$

By convention, 0 is encoded as $+0.10^0$. All encodings are possible representations of the triplets $(s, m, e)$. In this paper, $e$ is restricted to $[-100, 100]$, and $m$ to $[100, 999]$.

In base N positional encoding, $s$ (the sign) and $e$ (the exponent) are encoded as unique tokens: `+` or `-` for $s$, and one token from `E-100` to `E100` for $e$. The mantissa, $m$, is encoded as the representation of $m$ in base N (e.g. binary representation if $N = 2$, decimal representation if $N = 10$), a sequence of $\lceil log_N(1000) \rceil$ tokens from `0` to `N-1`. Overall, a number will be encoded as a sequence of $\lceil log_N(1000) \rceil + 2$ tokens, from a vocabulary of $202 + N$ tokens.

For instance, $x = e^\pi \approx 23.14069$, will be represented by $+231.10^{-1}$, and encoded in P10 (base 10 positional) as the sequence $[$`+,2,3,1,E-1`$]$, and in P1000 (base 1000 positional) as $[$`+,231,E-1`$]$. $x = -0.5$ will be represented as $-500.10^{-3}$, and encoded in P10 as $[$`-,5,0,0,E-3`$]$, and in P1000 as $[$`-,500,E-3`$]$. Other bases N could be considered, as well as different bases for the exponent, and different lengths for the mantissa. This paper uses two positional encodings: P10, which encodes numbers rounded to three significant digits, with absolute value in $[10^{-100}, 10^{101}]$, as sequences of 5 tokens, using a vocabulary of 213 tokens (10 digits, 2 signs, and 202 values of the exponent), and P1000 which encodes numbers as sequences of 3 tokens, with a vocabulary of 1104.

Balanced base $2a + 1$ uses digits between $-a$ and $a$ (Knuth, 1997). For instance, in balanced base 11, digits range from $-5$ to $5$. An every day example of a balanced base can be found in the way we state the hour as "twenty to two", or "twenty past two". Setting $a$ to 999 defines B1999, which encodes the sign and mantissa as a single token between $-999$ and $999$, and the exponent as in P10 and P1000. Numbers are encoded on two tokens, with a vocabulary of 2004.

For an even more compact representation, floating point numbers can be encoded as unique tokens, by rewriting any number as $x = m10^b$, with $m \in [-999, 999]$, $b \in [-(p+2)/2, (p+2)/2]$ and $p + 2 = 0, [2]$, and representing it as the unique token `FPm,b`. This allows to represent numbers with 3 significant digits and a dynamic range of $10^{p+2}$, using a vocabulary of $1800(p + 3)$ tokens. Setting $p = 14$, this paper introduces FP15, which encodes numbers as unique tokens with a vocabulary of $30,000$.

## B    $L^1$, $L^2$ and $L^\infty$ norms for evaluation

The accuracy of trained models is evaluated by decoding their predictions and verifying that they approximate the correct solution up to a fixed tolerance $\tau$. In the general case, if the model predicts a sequence $S_P$, and the solution of the problem is $O$, the prediction is considered to be correct if $S_P$ can be decoded into a matrix $P$ and

$$\|P - O\| < \tau \|O\| \tag{1}$$

For eigenvalue decomposition, the solution is correct if it can be decomposed as a pair $(Q, D)$ that verifies: $\|Q^T IQ - D\| < \tau \|D\|$ ($I$ the input matrix), i.e. that $Q$ diagonalizes $I$ into $D$. For singular value decomposition, the solution must verify $\|UIV - S\| < \tau \|S\|$, and for matrix inversion $\|PI - Id\| < \tau \|Id\| = \tau$. The matrix norm $L^1$: $\|A\| = \sum_{i,j} |a_{i,j}|$, for $A = (a_{i,j})$ is used throughout this paper. This section discusses its advantage over two other possible norms: $L^2$ ($\|A\| = \sum_{i,j} a_{i,j}^2$), and $L^\infty$ ($\|A\| = \max_{i,j} |a_{i,j}|$).

Using $L^1$ norm in equation 1 amounts to comparing the average absolute error on the predicted coefficients $(P - O)$ to the average absolute value of coefficients of $O$. $L^2$ compares the squared values and errors, and $L^\infty$ the largest absolute error to the largest coefficient in $|O|$. Compared to $L^1$, $L^2$ and $L^\infty$ (Max) emphasize large absolute errors, and large absolute coefficients of $O$. The impact of the norm varies from one problem to another. Figure 1 presents learning curves using the three norms for our best models, on different problems.

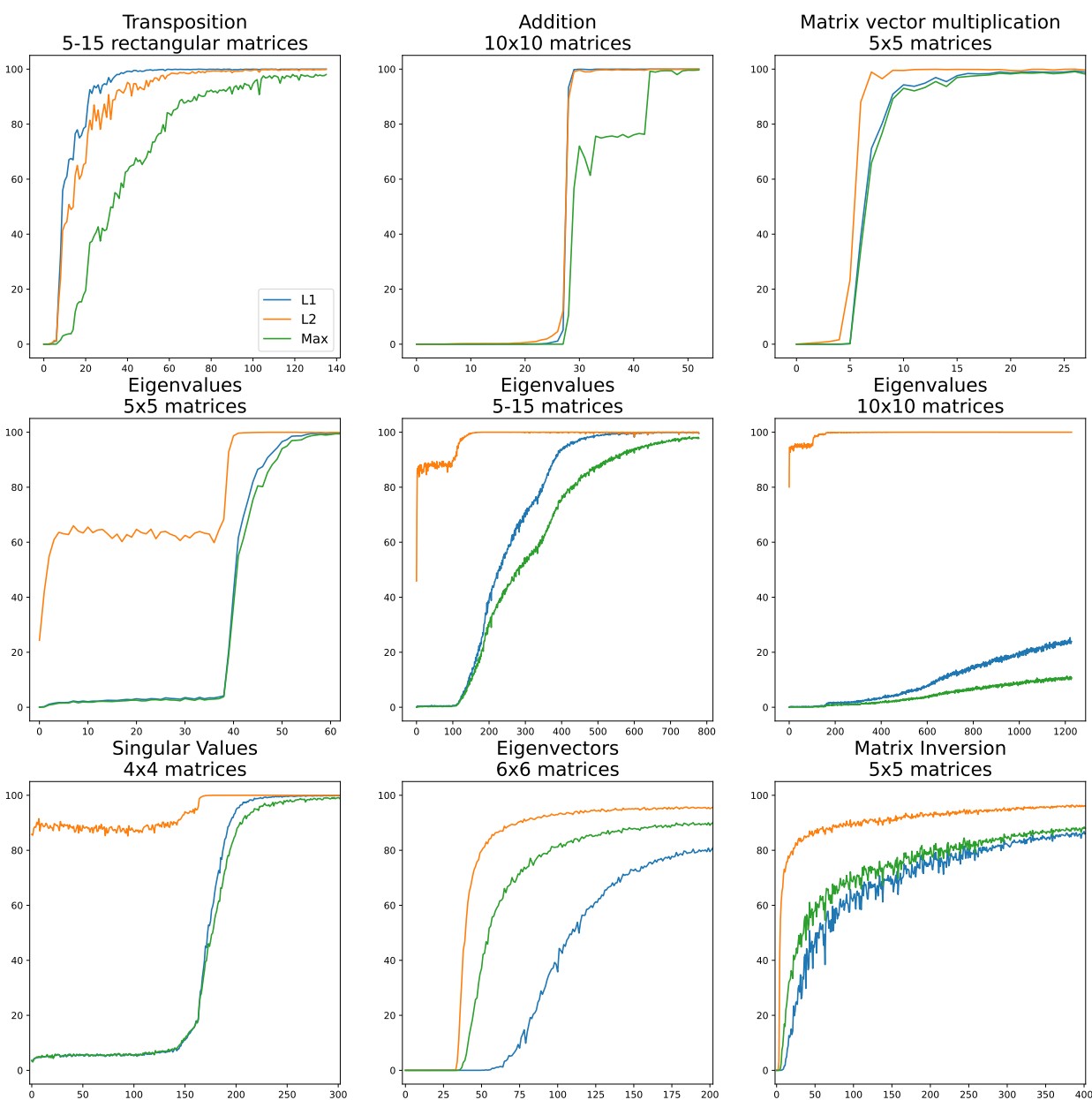

Figure 1: **Learning accuracies for different problems measured with norms $L^1$, $L^2$ and $L^\infty$ (Max).**

For basic arithmetic operations (transposition, addition, multiplication), there is little difference between $L^1$ and $L^2$ accuracies, and no reason to prefer one over the other for model evaluation. $L^\infty$ provides a stricter criterion for accuracy, but it has little practical impact.

For eigenvalue and singular value problems, $L^2$ accuracies reach a high value early during training, long before the model begins to learn according to the other norms. This is due to the fact that the eigenvalues of Wigner matrices tend to be regularly spaced over the interval $[-2\sigma, 2\sigma]$ ($\sigma = \sqrt{n}s$ with $s$ the standard deviation of coefficients and $n$ the dimension of the matrix). This means that the model can predict the largest absolute eigenvalues from the distribution of the coefficients, which can be computed from the dataset. For this reason, $L^2$ accuracy is not a good evaluation metric for the eigenvalue or singular value problem. This is particularly clear in the $10 \times 10$ case: transformers struggle with such matrices, and $L^1$ and $L^\infty$

accuracies remain very low even after a thousand epochs (300 million examples), but $L^2$ accuracy is close to 100% since the beginning of training.

A similar phenomenon takes place for eigenvector calculations: $L^2$ and $L^\infty$ accuracy rise steeply, long before the model begins to learn according to the $L^1$ norm. In this task, the model predicts both the eigenvalues and the coefficients of the matrix of eigenvectors $Q$. Because $Q$ is orthogonal, its coefficients will usually have small absolute values, compared to those of eigenvalues. As training goes on, the largest eigenvalue is first predicted, which causes the rise in the $L^2$ curve, then other eigenvalues are, which cause the rise in the $L^\infty$ curve, and finally the eigenvectors are correctly predicted, which is depicted in the (much slower) rise of the $L^1$ curve. Again, using $L^2$ or $L^\infty$ amounts to evaluating an easier problem (computing eigenvalues) than the one we are currently solving (eigen decomposition). These observations motivate the choice of $L^1$ as our evaluation norm.

## C  Impact of number precision

In all experiments, matrix coefficients are rounded to three significant digits. Three-digit precision was selected in order to keep the size of the FP15 vocabulary manageable. With three-digit precision, FP15 uses a vocabulary of 30 000 words, with four digits, it would use 300 000 words, and would be difficult to train on the small transformers this paper focuses on.

In this section, I investigate the impact of number precision on $10 \times 10$ matrix addition and $5 \times 5$ eigenvalue computation. Random matrices are rounded to two, three and four significant digits, using the P100, P1000 and P10000 encoding (i.e. numbers encoded on three tokens, mantissa in base 100, 1000 and 10000). I train transformers with 512 dimensions, 8 attention heads, and 2/2 layers (addition) or 4/1 layers (eigenvalues). Results for 10, 5, 2, 1, 0.5 and 0.1% tolerance are presented in tables 13 and 14.

On the addition task, rounding precision has no impact for tolerances larger than 1%: all models achieve close to 100% accuracy. At 0.5 tolerance, models trained with 2-digit precision are penalised by rounding error, but there is no significant difference between 3 and 4-digit precision. However, 4-digit models need significantly more examples to train: whereas 2 and 3-digit models achieve 99% accuracy at 5% tolerance after 5 million examples, a 4-digit model need 21 millions to reach 99% accuracy.

| Tolerance | 10 | 5 | 2 | 1 | 0.5 | 0.1 |
|---|---|---|---|---|---|---|
| 2-digit precision, P100 | 100 | 100 | 100 | 99.4 | 60.8 | 0 |
| 3-digit precision, P1000 | 100 | 100 | 99.3 | 98.1 | 97.2 | 66.4 |
| 4-digit precision, P10000 | 99.5 | 99.4 | 99.4 | 99.0 | 98.8 | 17.3 |

Table 13: **Accuracy of** $10 \times 10$ **matrix addition, for different precision and tolerance,** after training on 30 million examples.

For eigenvalues, all models achieve 100% accuracy at 5% tolerance. At lower tolerance (1 or 0.5%), accuracy increases with precision. On this task, learning speed is comparable for all models: 2, 3 and 4-digit models achieve 99% accuracy (with 5% tolerance) after 9, 8 and 7 million examples respectively. Overall, number precision has a marginal effect on accuracy.

| Tolerance | 10 | 5 | 2 | 1 | 0.5 | 0.1 |
|---|---|---|---|---|---|---|
| 2-digit precision, P100 | 100 | 100 | 94.3 | 87.4 | 80.1 | 24.1 |
| 3-digit precision, P1000 | 100 | 100 | 99.9 | 98.2 | 78.9 | 9.8 |
| 4-digit precision, P10000 | 100 | 100 | 99.9 | 99.0 | 85.6 | 1.4 |

Table 14: **Accuracy of** $5 \times 5$ **eigenvalue calculation, for different precision and tolerance,** after training on 60 million examples.

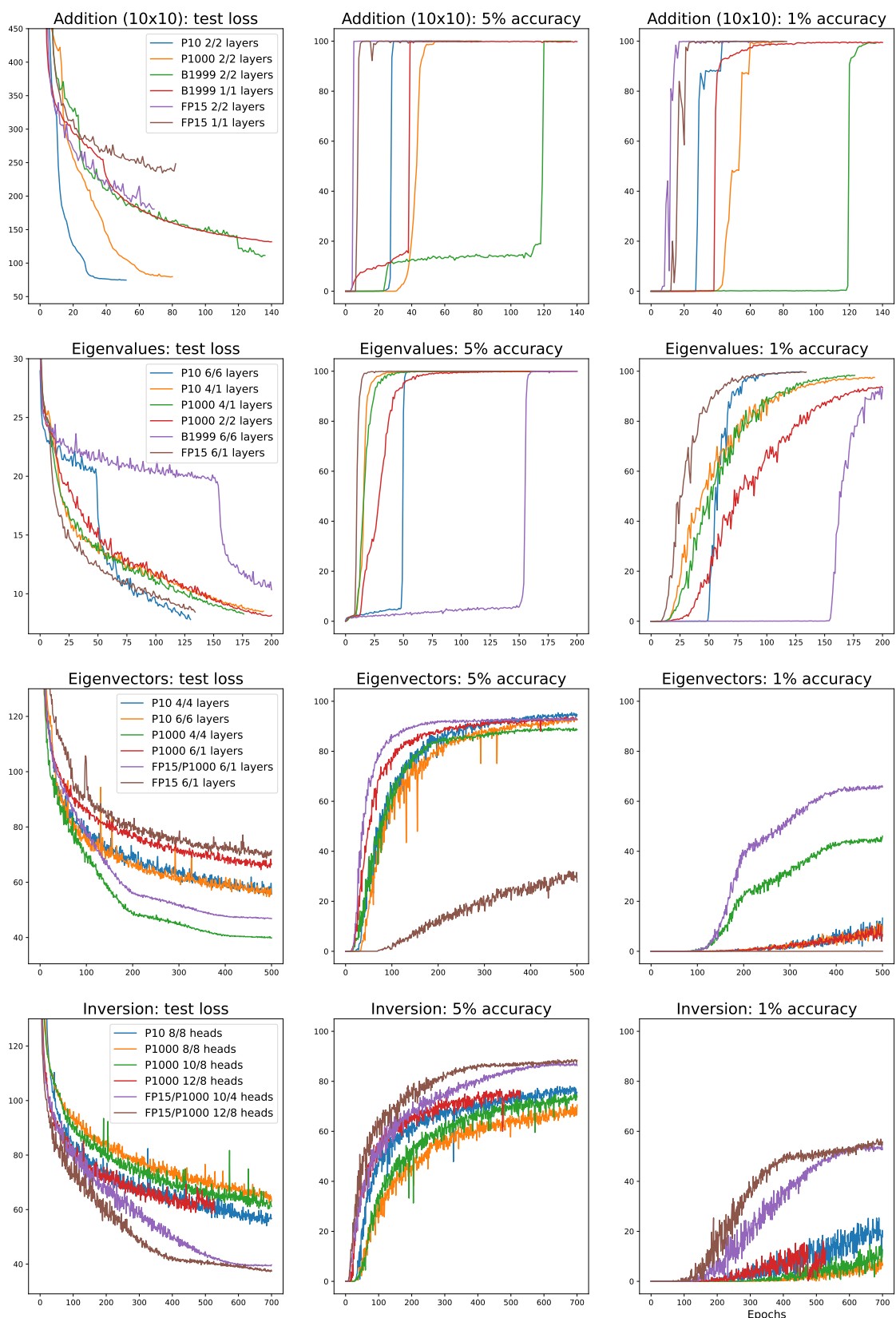

Figure 2: **Learning curves for different problems.** All problems except addition use $5 \times 5$ matrices. All models have 512 dimensions and 8/8 heads (except when mentioned in the legend). Inversion models have 6/1 layers. Epochs correspond to 300,000 training examples. Test loss is cross-entropy.

## D   Additional experimental results

### D.1   Learning curves for different encodings and architectures

Figure 2 presents learning curves (test loss, 5 and 1% accuracy) for ($10 \times 10$) addition, and ($5 \times 5$eigenvalues, eigenvectors and inversion. The best models learn to perform addition (1% tolerance) in less than 10 epochs (3 million examples). For other tasks, training size increases with operation complexity: from 20 million (70 epochs) for eigenvalues, to 50 million for eigenvectors, and over 120 million for matrix inversion. Some of the learning curves exhibit the "step shapes" often observed in arithmetic tasks: losses are subject to brutal drops, corresponding to steep increases in accuracy.

The learning curves for the harder problems (eigenvalues, eigenvectors and inversion) are noisy. This is caused by the learning rates: our models usually need small learning rates ($5 \times 10^{-4}$ before scheduling is typical) and there is a trade-off between low rates that will stabilize the learning curve, and larger rates that accelerate training.

### D.2   Impact of model size on accuracy and learning speed

Two main factors influence model size: the number of layers and the number of dimensions (see Appendix G for precise calculations). This section discusses their impact on accuracy and learning speed, when adding $10 \times 10$ matrices, multiplying a $5 \times 5$ matrix by a vector, and computing the eigenvalues of a $5 \times 5$ matrix. All the models in this section are symmetric (same dimension and number of layers in the encoder and decoder) and have 8 attention heads.

For the addition task, tables 15 and 16 present the accuracy after 60 epochs (18 million examples) and the number of epochs (of 300,000 examples) needed to reach 95% accuracy, for models using the P1000 and B1999 encoding. Shallow architectures (i.e. 1 or 2 layers) learn addition with high accuracy for both encodings, but the more compact B1999 supports smaller models (256 dimensions). In terms of speed, shallow (2-layer) B1999 and deep (6-layer) P100 prove fastest.

| | B1999 | | | | P1000 | | | |
|---|---|---|---|---|---|---|---|---|
| dimension | 64 | 128 | 256 | 512 | 64 | 128 | 256 | 512 |
| 1/1 layers | 31 | 7 | 82 | 100 | 0 | 0 | 1 | 40 |
| 2/2 layers | 0 | 0 | 100 | 100 | 0 | 0 | 0 | 99 |
| 4/4 layers | 0 | 0 | 0 | 14 | 0 | 0 | 0 | 98 |
| 6/6 layers | 0 | 0 | 0 | 0 | 0 | 0 | 0 | 99 |

Table 15: **Accuracy of matrix addition for different model sizes.** $10 \times 10$ matrices, 60 epochs (18 millions examples), 5% tolerance.

| | B1999 | | | | P1000 | | | |
|---|---|---|---|---|---|---|---|---|
| dimension | 64 | 128 | 256 | 512 | 64 | 128 | 256 | 512 |
| 1/1 layers | - | - | 76 | 15 | - | - | - | 96 |
| 2/2 layers | - | - | 26 | 6 | - | - | - | 37 |
| 4/4 layers | - | - | 70 | 63 | - | - | - | 53 |
| 6/6 layers | - | - | - | - | - | - | - | 23 |

Table 16: **Learning speed of matrix addition for different model sizes.** Number of epochs needed to reach 95% accuracy (5% tolerance). 1 epoch = 300,000 examples.

Table 17 presents the learning speed of models of different sizes for the matrix/vector product and eigenvalue computation tasks ($5 \times 5$ matrices, and P1000 encoding). For each problem, there exists a minimal dimension and depth under which models struggle to learn: one layer and 128 dimensions for products, one layer or 128 dimensions for eigenvalues. Over that limit, increasing the dimension accelerates learning. Increasing the depth, on the other hand, bring no clear improvement in speed or accuracy.

| | Matrix product | | | Eigenvalues | | | |
|---|---|---|---|---|---|---|---|
| | 128 | 256 | 512 | 128 | 256 | 512 | 1024 |
| 1/1 layers | - | 29 | 18 | - | - | - | - |
| 2/2 layers | 24 | 12 | 7 | - | 102 | 36 | 23 |
| 4/4 layers | 28 | 11 | 5 | 244 | 90 | 24 | 13 |
| 6/6 layers | 24 | 10 | 6 | - | - | 129 | 16 |
| 8/8 layers | 18 | 12 | 6 | - | - | 34 | 24 |

Table 17: **Learning speed of matrix and vector products and eigenvalue calculations for different model sizes.** Number of epochs needed to reach 95% accuracy (with 5% tolerance). 1 epoch = 300,000 examples. $5 \times 5$ matrices, P1000 encoding.

### D.3 Scaling to larger models

This paper mostly focuses on small models, with up to 6 layers and 50 million parameters. In this section, I investigate the performance of larger models, with up to 24 layers and over 400 million parameters. All models are asymmetric, featuring either a deep encoder and a shallow (1-layer) decoder, or the reverse. Shallow models have one layer, 512 dimensions and 8 attention heads. Deep models range from small (6 layers) to medium (8 layers), large (12 layers) and extra-large (24 layers). By default, they have a dimension of $512, 640, 768$ and $1024$, and $8, 10, 12$ and $16$ heads respectively. For each size, I also experiment with models with 50% more dimensions and attention heads. The basic encoding is FP15/P1000, but I also experiment with two alternative configurations: FP15/FP15 and P1000/P1000. Table 18 summarizes the configurations that were tested.

| Model size | Base configuration | Larger dimension | More heads |
|---|---|---|---|
| Small: 6 layers | 512 dimensions, 8 heads | 768 | 12 |
| Medium: 8 layers | 640 dimensions, 10 heads | 960 | 15 |
| Large: 12 layers | 768 dimensions, 12 heads | 1152 | 18 |
| Extra-large: 24 layers | 1024 dimensions, 16 heads | 1536 | 24 |

Table 18: **Large model configurations.** All three configurations are tested for the encoder and decoder, and with the three encodings: FP15/P1000, P1000/P1000, FP15/FP15.

For the eigenvalue task, models are trained on $8 \times 8$, $10 \times 10$ and $12 \times 12$ matrices. Table 19 presents the best performing configurations, for different tolerance levels, and the sample size needed to achieve 99% accuracy (with 5% tolerance). In general, architectures combining a deep decoder with a shallow encoder outperform deep encoders and shallow decoders. Deeper models tend to learn faster, and solve larger problems, but for a given number of layers, there is not clear advantage of increasing dimension or the number of attention heads. In particular, models with 12 or 24 layers can compute the eigenvalues of $12 \times 12$ matrices, a task inaccessible to small architectures. Learning to predict the eigenvalues of $8 \times 8$ matrices takes over 50 million examples for 6-layer models, but only 11 million for 24-layer architectures. For $10 \times 10$ matrices, 8-layer models need about 85 million examples, whereas 12 and 24-layer models need 38 and 36. Finally, it is interesting to note that larger models achieve better precision: 12-layer models routinely achieve accuracies over 90% with 0.5% tolerance, while smaller models struggle at such precision levels.

| Model | Dimensions | 5% | 2% | 1% | 0.5% | Sample size* |
|---|---|---|---|---|---|---|
| Small (6 layers) | | | | | | |
| E:1/512/8/FP15 D:6/516/**12**/P1000 | 8x8 | 100 | 99.5 | 88.8 | 35.9 | 51 |
| E:1/512/8/FP15 D:6/512/8/P1000 | 8x8 | 100 | 97.7 | 86.5 | 50.3 | 64.2 |
| **E:6**/**768**/8/P1000 **D:1**/512/8/P1000 | 8x8 | 100 | 98.8 | 80.8 | 28.1 | 64.8 |
| E:1/512/8/FP15 D:6/**768**/8/P1000 | 8x8 | 100 | 97.9 | 69.8 | 17.7 | 71.7 |
| E:1/512/8/FP15 D:6/516/**12**/P1000 | 10x10 | 99.7 | 72.6 | 22.3 | 2.2 | 194.7 |
| E:1/512/8/FP15 D:6/**768**/8/**FP15** | 10x10 | 83.6 | 17.0 | 1.5 | 0.1 | - |
| Medium (8 layers) | | | | | | |
| **E:8**/640/10/P1000 **D:1**/512/8/P1000 | 8x8 | 99.4 | 99.0 | 97.0 | 60.1 | 120.9 |
| E:1/512/8/FP15 D:8/645/**15**/P1000 | 8x8 | 100 | 99.9 | 95.9 | 56.1 | 45 |
| E:1/512/8/FP15 D:8/**960**/10/P1000 | 8x8 | 100 | 99.9 | 95.4 | 55.9 | 34.8 |
| E:1/512/8/FP15 D:8/640/10/P1000 | 8x8 | 100 | 99.8 | 92.0 | 42.0 | 43.8 |
| E:1/512/8/FP15 D:8/640/10/P1000 | 10x10 | 100 | 98.4 | 70.2 | 11.2 | 85.2 |
| E:1/512/8/FP15 D:8/645/**15**/P1000 | 10x10 | 99.1 | 62.9 | 14.1 | 1.0 | 108.6 |
| E:1/512/8/FP15 D:8/645/**15**/**FP15** | 12x12 | 3.0 | 0 | 0 | 0 | - |
| Large (12 layers) | | | | | | |
| E:1/512/8/FP15 D:12/**1152**/12/P1000 | 8x8 | 100 | 100 | **100** | **99.1** | 13.2 |
| E:1/512/8/FP15 D:12/768/12/P1000 | 8x8 | 100 | 100 | 99.9 | 93.5 | 42 |
| E:1/512/8/FP15 D:12/768/12/P1000 | 10x10 | 100 | 100 | **99.8** | 89.3 | 68.7 |
| E:1/512/8/FP15 D:12/**1152**/12/P1000 | 10x10 | 100 | 100 | 99.7 | **96.8** | 37.5 |
| E:1/512/8/FP15 D:12/768/12/P1000 | 12x12 | 96.7 | 24.2 | 1.5 | 0.1 | - |
| E:1/512/8/FP15 D:12/768/12/**FP15** | 12x12 | 85.2 | 11.8 | 0.6 | 0 | - |
| Extra-large (24 layers) | | | | | | |
| E:1/512/8/FP15 D:24/1024/16/P1000 | 8x8 | 100 | 100 | **100** | 99.2 | **11.1** |
| E:1/512/8/FP15 D:24/1032/**24**/P1000 | 8x8 | 100 | 100 | 100 | 93.9 | 14.1 |
| E:1/512/8/FP15 D:24/1032/**24**/**FP15** | 8x8 | 100 | 100 | 99.7 | 82.6 | 19.5 |
| E:1/512/8/FP15 D:24/1024/16/P1000 | 10x10 | 100 | 100 | 99.6 | 75.9 | **36.3** |
| E:1/512/8/FP15 D:24/1032/**24**/P1000 | 10x10 | 100 | 100 | 99.5 | 69.3 | 42.6 |
| E:1/512/8/FP15 D:24/1032/**24**/P1000 | 12x12 | 100 | 99.6 | **95.5** | **43.7** | 99.6 |
| E:1/512/8/FP15 D:24/1024/16/P1000 | 12x12 | 100 | 99.6 | 88.9 | 23.0 | **99.3** |

Table 19: **Large models: accuracy of eigenvalue computations.** E=Encoder, D=Decoder, 1/512/8 : 1 later, 512 dimensions, 8 heads. Sorted by 1% accuracy. *Millions of examples to reach 99% accuracy

## D.4 Model performance on different training sets

The models presented in the main part of this paper are trained on Wigner matrices (matrices with independent and identically distributed, iid, coefficients) with fixed-range coefficient. In section 5, I argued that different training sets allowed for better out-of-domain generalization. Table 20 summarizes in-domain performance (i.e. accuracy when the test set is has the same distribution as the training set) on different training sets.

Wigner matrices with uniform or Gaussian distributed, and fixed or variable-range, coefficients are learned to high accuracy (more than 99%) by all models. The eigenvalues of non-Wigner matrices with Gaussian or Laplace distributed eigenvalues, and of mixtures of Wigner and non-Wigner matrices, are also predicted to high accuracy by all models. Over matrices with positive or uniformly distributed eigenvalues, smaller models using the FP15 encoding prove difficult to train.

|  | FP15 | | P1000 | |
|---|---|---|---|---|
|  | 4/1 layers | 6/1 layers | 4/1 layers | 6/1 layers |
| **Wigner matrices (iid coefficients)** | | | | |
| Uniform iid A=10 | 99.6 | 100 | 99.8 | 100 |
| Gaussian iid A=10 | 99.8 | 100 | 99.8 | 100 |
| Uniform iid A=1,100 | 99.0 | 99.2 | 99.8 | 100 |
| Uniform iid A=1,1000 | 99.2 | 99.5 | 99.7 | 99.8 |
| **Non Wigner** | | | | |
| Positive A=10 | 12.7 | 100 | 100 | 100 |
| Uniform A=10 | 8.2 | 10.8 | 99.9 | 100 |
| Gaussian A=10 | 99.6 | 100 | 100 | 100 |
| Laplace A=10 | 99.4 | 99.9 | 99.9 | 99.9 |
| Gaussian+uniform+Laplace A=10 | 3.8 | 99.8 | 99.6 | 99.9 |
| **Wigner and non-Wigner mixtures** | | | | |
| iid+gaussian A=10 | 99.5 | 99.9 | 98.0 | 99.7 |
| iid+positive A=10 | 99.8 | 99.9 | 99.8 | 99.8 |
| iid+Laplace A=10 | 99.6 | 99.8 | 99.6 | 99.5 |
| iid+positive+gaussian A=10 | 99.8 | 99.9 | 99.7 | 99.9 |
| iid+positive+Laplace A=10 | 99.0 | 99.8 | 99.6 | 99.8 |

Table 20: **In-distribution eigenvalue accuracy (tolerance 2%) for different training distributions.** All models have 512 dimensions, and 8 attention heads, and are trained on 5x5 matrices.

# E  Alternative architectures

## E.1  Other sequence to sequence models : LSTM and GRU

I experimented with two popular recurrent architectures: long short-term memories (LSTM Hochreiter & Schmidhuber (1997)), and gated recurrent units (GRU Cho et al. (2014)), on three tasks : addition of $5 \times 5$ and $10 \times 10$ matrices, eigenvalues and matrix inversion of $5 \times 5$ matrices. All models are sequence-to-sequence architectures: an encoder and a decoder (LSTM or GRU), with 2 to 8 layers, and 1024 or 2048 hidden dimensions. The input and output sequences, encoded as in the rest of the paper, are pre-processed (and decoded) via an embedding layer with 256 or 512 dimensions.

Addition, a very easy task for transformers (see section 4.2) proves difficult for LSTM and GRU. None of the models can learn addition of $10 \times 10$ matrices. Some models can learn addition of $5 \times 5$ matrices, but whereas transformers achieve 100% accuracy for all tolerances, the best LSTM and GRU only exceed 90% at 1% tolerance. GRU seem to perform better than LSTM on this task, and 2-layer models perform better than 4-layer models, but transformers have a distinct advantage over LSTM and GRU for addition.

Both LSTM and GRU can be trained to predict eigenvalues of $5 \times 5$ matrices with the same accuracy as transformers, for the P1000 and FP15 encoding (table 22). Matrix inversion, on the other hand, cannot be learned. Overall, these experiments show that other sequence to sequence architectures, LSTM and GRU, can learn tasks like eigenvalues and addition of small matrices. However, they are less efficient on addition (in terms of precision and scaling to larger matrices) and fail on more complex tasks, like matrix inversion.

## E.2  Shared-layer transformers: Universal transformers

In the Universal Transformer (Dehghani et al., 2018), the stacked layers of usual transformer implementations are replaced by one layer that is looped through a fixed number of times (feeding the output of one iteration into the input of the next). This amounts to sharing the weights of the different layers, therefore greatly reducing the number of parameters in the model. This technique can be applied to the encoder, the decoder or both. The number of iterations is a fixed hyperparameter, but the original paper also proposed a halting mechanism inspired by Adaptive Computation Time (Graves, 2016), to adaptively control loop length at the token level. In this version, a stopping probability is learned for every position in the input sequence, and

|  | 2 layers | | | | 4 layers | | | |
| --- | --- | --- | --- | --- | --- | --- | --- | --- |
| Hidden dimension | 1024 | | 2048 | | 1024 | | 2048 | |
| Embedding dimension | 256 | 512 | 256 | 512 | 256 | 512 | 256 | 512 |
| **Long short-term memory** | | | | | | | | |
| 5% tolerance | 100 | 0 | 0 | 100 | 0 | 0 | 0 | 0 |
| 2% tolerance | 98 | 0 | 0 | 100 | 0 | 0 | 0 | 0 |
| 1% tolerance | 95 | 0 | 0 | 86 | 0 | 0 | 0 | 0 |
| 0.5% tolerance | 34 | 0 | 0 | 1 | 0 | 0 | 0 | 0 |
| **Gated recurrent Units** | | | | | | | | |
| 5% tolerance | 100 | 100 | 0 | 100 | 0 | 100 | 0 | 0 |
| 2% tolerance | 100 | 28 | 0 | 100 | 0 | 99 | 0 | 0 |
| 1% tolerance | 44 | 0 | 0 | 91 | 0 | 74 | 0 | 0 |
| 0.5% tolerance | 0 | 0 | 0 | 9 | 0 | 4 | 0 | 0 |

Table 21: $5 \times 5$ **matrix addition using LSTM and GRU.**

|  | FP15 | | | | | | P1000 | | | | | |
| --- | --- | --- | --- | --- | --- | --- | --- | --- | --- | --- | --- | --- |
| Hidden dimension | 1024 | | | 2048 | | | 1024 | | | 2048 | | |
| Layers | 4 | 6 | 8 | 4 | 6 | 8 | 4 | 6 | 8 | 4 | 6 | 8 |
| **LSTM** | | | | | | | | | | | | |
| 5% tolerance | 100 | 100 | 100 | 100 | 100 | 6 | 100 | 100 | 5 | 100 | 100 | 100 |
| 2% tolerance | 95 | 100 | 100 | 99 | 100 | 1 | 100 | 100 | 1 | 100 | 99 | 100 |
| 1% tolerance | 78 | 98 | 99 | 91 | 98 | 0 | 97 | 98 | 0 | 100 | 92 | 99 |
| 0.5% tolerance | 46 | 81 | 83 | 62 | 68 | 0 | 78 | 88 | 0 | 89 | 57 | 76 |
| **Gated recurrent Units** | | | | | | | | | | | | |
| 5% tolerance | 100 | 100 | 100 | 100 | 100 | 100 | 100 | 100 | 100 | 100 | 5 | 100 |
| 2% tolerance | 98 | 99 | 100 | 100 | 100 | 100 | 99 | 100 | 100 | 100 | 1 | 100 |
| 1% tolerance | 86 | 93 | 96 | 98 | 99 | 97 | 94 | 98 | 95 | 97 | 0 | 98 |
| 0.5% tolerance | 53 | 68 | 75 | 78 | 83 | 65 | 65 | 76 | 63 | 75 | 0 | 66 |

Table 22: **Eigenvalue computation with LSTM and GRU,** $5 \times 5$ matrices.

once it reaches a certain threshold, the layer merely copies the input onto the output. The iteration stops when all positions have halted, or a specific value is reached. A recent paper (Csordás et al., 2021) proposes to use a similar copy-gating mechanism to skip iterations in a fixed-length loop. I experiment with these three variants (fixed length, adaptive halting, copy gating) on the addition (of $10 \times 10$ matrices), eigenvalue and matrix inversion tasks ($5 \times 5$ matrices).

For the addition task, I train universal transformers with one layer and in the encoder and decoder, 256 or 512 dimensions and 8 attention heads, and use the B1999 encoding for the data. I experiment with looped encoder, looped decoder, and loop in both, a loop length of 4, copy-gating and ACT (the 4 loops in then a maximum number of iterations)and copy-gating. Table 23 summarizes my findings. Only models with encoder loops learn to add, and models with 512 dimensions learn with over 95% accuracy for all tolerances. Universal Transformers with one layer (looped-encoder only) perform as well as 2/2 transformers.

On the eigenvalue task, I experiment on the P1000 and FP15 encoding, with encoder-loop only 1/1 Universal Transformers with 4 or 8 loops. Universal transformers using the P1000 encoding achieve the same performances (with only one layer) than transformers from section 4.4. 4 loop transformers seem to perform best, gating does not seem to improve performance and ACT slightly degrades it. With the FP15 encoding, universal transformers become very difficult to train: only the 4 loop gated version achieves significant accuracy (still lower than the 6/1 transformers).

Finally, I experimented with matrix inversion, with FP15/P1000 and P1000/P1000 encodings, and 4 or 8 loops in the encoder. A gated universal transformer using FP15 in the input and P1000 in the output achieved 73% accuracy, a significant result albeit lower than the best result achieved with 6/1 transformers using the same encodings (90%). With the P1000 encoding, the best universal transformers reach 55% accuracy,

|  | 5% | 2% | 1% | 0.5% |
|---|---|---|---|---|
| Looped encoder | | | | |
| 256 dimensions, 4 loops | 15 | 1 | 0 | 0 |
| 512 dimensions, 4 loops | 100 | 100 | 100 | 100 |
| 256 dimensions, 4 loops, gated | 97 | 66 | 41 | 29 |
| 512 dimensions, 4 loops, gated | 100 | 100 | 100 | 100 |
| 256 dimensions, 4 loops, ACT | 100 | 92 | 76 | 66 |
| 512 dimensions, 4 loops, ACT | 100 | 100 | 98 | 96 |
| Looped decoder | 0 | 0 | 0 | 0 |
| Looped encoder and decoder | 0 | 0 | 0 | 0 |
| 2/2 transformer (baseline) | 100 | 100 | 100 | 100 |

Table 23: **Accuracy of Universal transformers**, $10 \times 10$ matrix addition for different tolerances.

|  | 5% | 2% | 1% | 0.5% |
|---|---|---|---|---|
| P1000 | | | | |
| 4 loops | 100 | 100 | 97 | 87 |
| 8 loops | 100 | 99 | 93 | 69 |
| 4 loops, gated | 100 | 100 | 98 | 91 |
| 8 loops, gated | 100 | 100 | 99 | 90 |
| 4 loops, ACT | 100 | 97 | 89 | 62 |
| 8 loops, ACT | 100 | 95 | 77 | 42 |
| FP15 | | | | |
| 4 loops | 4 | 0 | 0 | 0 |
| 8 loops | 0 | 0 | 0 | 0 |
| 4 loops, gated | 94 | 84 | 57 | 23 |
| 8 loops, gated | 6 | 1 | 0 | 0 |
| 4 loops, ACT | 4 | 0 | 0 | 0 |
| 8 loops, ACT | 4 | 0 | 0 | 0 |
| 4/1 transformer (P1000 baseline) | 100 | 100 | 99 | 89 |
| 6/1 transformer (FP15 baseline) | 100 | 100 | 100 | 92 |

Table 24: **Accuracy of Universal transformers,** $5 \times 5$ matrices eigenvalue computation for different tolerances.

compared to 80% for their 6/1 transformer counterparts. Overall, Universal Transformers seem to achieve comparable performances with deep transformers (except on the inversion tasks), using less parameters. This makes shared layer transformers an interesting direction for future work.

# F   Additional experiments

## F.1   Retraining

Models trained on matrices of a given size do not generalize to different dimensions, but they can be retrained over samples of matrices of different size. This takes comparatively few examples: a $5 \times 5$ model, that takes 40 million examples to be trained, can learn to predict with high accuracy eigenvalues of matrices of dimension 6 and 7 with about 25 million additional examples. Table 25 presents those results. The possibility to retrain large transformers (such as GPT-3) on different tasks is well documented, it is interesting to observe the same phenomenon in smaller models.

| Encoding | Retrain dimensions | Accuracy (5%) | Accuracy (2%) | Retrain examples |
|----------|--------------------|--------------:|--------------:|------------------|
| P10      | 5-6                | 100           | 99.9          | 10M              |
| P10      | 5-7                | 100           | 93.6          | 25M              |
| P1000    | 5-6                | 100           | 97.7          | 25M              |

Table 25: **Model accuracy after retraining.** Models trained over 5x5 matrices, retrained over 5-6 and 5-7. Overall performance after retraining (tolerance 5 and 2%), and number of examples needed for retraining. All models have 512 dimensions and 8 attention heads

## F.2  Joint training: learning to perform several operations

All models so far are trained on just one task. In this section, I investigate joint learning: training one model to perform several operations. To this effect, a token is added at the beginning of the input and output sequence, to indicate the task (e.g. `Transpose` or `Add`), and generate training data by randomly mixing examples of the different operations to be performed.

Transformers with 4 or 6 layers, 512 dimensions and 8 attention heads are trained on eight datasets corresponding to the following joint training goals (all operations in equal proportions):

- Transpose and add (TA)
- Transpose, add and dot product (vector matrix multiplication) (TAD)
- Transpose, add, dot product and matrix multiplication (TADM)
- Transpose, add, dot product, matrix multiplication and eigenvalues (TADME)
- Transpose, add, dot product, matrix multiplication, eigenvalues and eigenvectors (TADMEF)
- Transpose, add, dot product, matrix multiplication, eigenvalues, eigenvectors and matrix inversion (TADMEFI)
- Eigenvalues, eigenvectors and matrix inversion (EFI)

Table 26 summarizes results.

|         | T   | A   | D   | M   | E   | F   | I   |
|---------|-----|-----|-----|-----|-----|-----|-----|
| TA      | 100 | 100 |     |     |     |     |     |
| TAD     | 100 | 100 | 100 |     |     |     |     |
| TADM    | 100 | 100 | 100 | 100 |     |     |     |
| TADME   | 100 | 100 | 26  | 100 | 80  |     |     |
| TADMEF  | 100 | 100 | 100 | 100 | 3   | 0   |     |
| TADMEFI | 100 | 100 | 100 | 100 | 3   | 0   | 0   |
| EFI     |     |     |     |     | 100 | 22  | 0   |

Table 26: **Accuracy of joint training,** $5 \times 5$ matrices, 5% tolerance.

Over mixtures of the four basic operations (transposition, addition, dot products and multiplication: goals TA, TAD and TADM), models predict all operations with almost perfect accuracy. Joint training on the basic operations and eigenvalue computations (the TADME task) allows the model to predict eigenvalues with 80% accuracy, in exchange for a loss of performances on the dot product task. As the number of non-basic tasks increases, the model keeps learning basic operations to 100% accuracy (as in the TADM setting), but the more advanced operations are not learned. Joint training on the advanced tasks only (eigenvalues, vectors and inversion) results in 100% accuracy on eigenvalue computation, 22% on eigenvectors, and 0 on inversion. These results demonstrate the feasibility of joint training on basic matrix operations, but also suggest that further research is needed if one wants to extend joint training to all the tasks considered in this paper.

## F.3  Additional results with noisy data

See tables 27 and 28.

| | B1999 | | | | P1000 | | | |
| --- | --- | --- | --- | --- | --- | --- | --- | --- |
| | 2/2 layers | | 4/4 layers | | 2/2 layers | | 4/4 layers | |
| | 256 | 512 | 256 | 512 | 256 | 512 | 256 | 512 |
| 5% tolerance | | | | | | | | |
| $0.01\sigma$ error | 100 | 100 | 100 | 100 | 100 | 100 | 99.4 | 100 |
| $0.02\sigma$ | 100 | 100 | 99.8 | 100 | 100 | 100 | 100 | 100 |
| $0.05\sigma$ | 41.5 | 41.2 | 41.7 | 41.6 | 39.3 | 41.2 | 39.4 | 40.7 |
| 2% tolerance | | | | | | | | |
| $0.01\sigma$ error | 99.8 | 99.9 | 99.8 | 99.9 | 99.4 | 100 | 98.2 | 99.9 |
| $0.02\sigma$ | 43.7 | 44.2 | 42.1 | 44.7 | 39.0 | 44.9 | 42.6 | 45.3 |
| $0.05\sigma$ | 0 | 0 | 0 | 0 | 0 | 0 | 0 | 0 |
| 1% tolerance | | | | | | | | |
| $0.01\sigma$ error | 39.8 | 41.7 | 39.6 | 44.0 | 36.6 | 44.0 | 28.9 | 44.6 |
| $0.02\sigma$ | 0.1 | 0.1 | 0.1 | 0.1 | 0.1 | 0.1 | 0.1 | 0.1 |
| $0.05\sigma$ | 0 | 0 | 0 | 0 | 0 | 0 | 0 | 0 |

Table 27: **Accuracy of noisy $5 \times 5$ matrix addition for different error levels and tolerances.**

| | FP15 | | | | P1000 | | | |
| --- | --- | --- | --- | --- | --- | --- | --- | --- |
| | 4/4 layers | | 6/6 layers | | 4/4 layers | | 6/6 layers | |
| | 512 | 1024 | 512 | 1024 | 512 | 1024 | 512 | 1024 |
| 5% tolerance | | | | | | | | |
| $0.01\sigma$ error | 6.1 | 100 | 5.1 | 6.0 | 100 | 100 | 100 | 100 |
| $0.02\sigma$ | 100 | 100 | 6.7 | 100 | 100 | 100 | 100 | 100 |
| $0.05\sigma$ | 99.1 | 99.3 | 99.3 | 6.4 | 99.3 | 99.0 | 99.0 | 98.8 |
| 2% tolerance | | | | | | | | |
| $0.01\sigma$ error | 0.7 | 99.8 | 0.5 | 0.8 | 99.3 | 99.6 | 99.9 | 99.8 |
| $0.02\sigma$ | 97.0 | 97.1 | 0.8 | 88.4 | 97.3 | 97.9 | 93.1 | 95.4 |
| $0.05\sigma$ | 37.9 | 38.4 | 40.6 | 0.5 | 40.1 | 37.3 | 37.5 | 35.3 |
| 1% tolerance | | | | | | | | |
| $0.01\sigma$ error | 0.1 | 82.1 | 0.1 | 0.2 | 79.7 | 83.8 | 87.9 | 83.8 |
| $0.02\sigma$ | 47.8 | 51.3 | 0.1 | 26.1 | 46.2 | 47.5 | 36.4 | 41.3 |
| $0.05\sigma$ | 3.8 | 4.2 | 4.1 | 0.1 | 4.1 | 3.8 | 3.9 | 3.4 |

Table 28: **Accuracy of noisy eigenvalue computations, for different error levels and tolerances, $5 \times 5$ matrices.**

## G   Number of parameters

The number of parameters in the sequence-to-sequence transformers used in this paper can be calculated as follows.

- A self-attention mechanism with dimension $d$ has $4d(d+1)$ parameters: it is composed of four linear layers (K, Q, V and the output layer), with $d$ input, $d$ output and a bias.
- A cross-attention mechanism with $d_e$ dimensions in the encoder, and $d$ in the decoder has $2d(d+d_e+2)$ parameters (K and V are $d_e \times d$ layers).
- A FFN with one hidden layer, $d$ input and output, and $h$ hidden units has $d(h+1) + h(d+1)$ parameters.
- A layer normalization with $d$ dimensions has $2d$ parameters.
- An encoder layer with dimension $d$ has a self-attention mechanism, a FFN with $4d$ hidden units (in this implementation) and two layer normalizations, for a total number of parameters of $12d^2 + 13d$.
- A decoder layer has a cross-attention layer (encoding dimension $d_e$) and a layer normalization on top of an encoder, for a total of $14d^2 + 19d + 2d_ed$ parameters.

- An embedding of dimension $d$ for a vocabulary of $w$ words will use $dw$ parameters, and $2d$ more if it is coupled to a layer normalization.
- The final prediction layer with an output dimension of $d$ and a decoded vocabulary of $w$ words will use $(d+1)w$ parameters (but in this case, $dw$ will be shared with the decoder embedding).

Overall, the number of parameters for a transformer with $n_e$ encoding layers with dimension $d_e$, $n*d$ decoding layers with dimension $d_d$, an input vocabulary of $w_i$ words, an output vocabulary of $w_o$ words and a positional embedding of $w_p$ words (corresponding to the maximum sequence length) can be computed by the formula:

$$P = d_e(w_i + w_p + 2) + ((w_o + w_p + 2)d_d + w_o) + n_e d_e(12d_e + 13) + n_d d_d(14d_d + 2d_e + 19)$$

the four terms in the sum corresponding to the input embedding, the output embedding, the encoder and the decoder.

Table 29 provides the number of parameters for some of the models used in this paper. For the positional embedding, the number of words is the longest input and output sequence studied with that model.

| Experiment | Model | Parameters |
|---|---|---|
| Transposition | 1/1 layers 256 dimensions P10 | 2,276,171 |
| | 1/1 layers 256 dimensions P1000 | 2,737,871 |
| | 1/1 layers 256 dimensions B1999 | 3,297,554 |
| | 1/1 layers 256 dimensions FP15 | 17,045,441 |
| Addition | 2/2 layers, 512 dimensions, B1999 | 17,619,218 |
| Matrix vector multiplication | 2/2 layers 512 dimensions P10 | 15,578,443 |
| | 2/2 layers 512 dimensions P1000 | 16,500,943 |
| | 4/4 layers 512 dimensions P1000 | 31,213,775 |
| Matrix multiplication | 1/4 layers 512 dimensions P1000 | 21,756,623 |
| | 1/6 layers 512 dimensions P1000 | 30,164,687 |
| Eigen decomposition | 1/6 layers 512 dimensions FP15 | 58,751,937 |
| | 6/1 layers 512 dimensions FP15 | 53,493,697 |
| | 6/1 layers 512 dimensions P1000 | 24,906,447 |
| | 6/6 layers 512 dimensions P1000 | 45,926,607 |
| Matrix inversion | 6/1 layers 512 dimensions FP15/P1000 | 39,186,127 |

Table 29: **Number of parameters of transformers used in the paper.**

## H    Eigenvalue distribution of Wigner matrices, an empirical justification

Figure 3 provides an empirical confirmation of the property of Wigner matrices mentioned in sections 2.2 and 5: the standard deviation of their eigenvalues is a function of their dimension and standard deviation of their coefficients only, and does not depend on the actual distribution of the coefficients. In particular, for coefficients with standard deviation $\sigma = 10/\sqrt{3} = 5.77$, we expect the standard deviation of their eigenvalue distribution to be $\sigma = 12.91, 18.26, 22.36$ and $25.81$ for square matrices of dimension $5, 10, 15$ and $20$.

For three distributions, uniform, Laplace and gaussian, and four dimensions ($5, 10, 15,$ and $20$), 100 000 random matrices with the same standard deviation of coefficients were generated, and their eigenvalues were computed. Standard deviations are within 0.01 of theoretical values for all distributions and dimensions. It is interesting to note how the distributions (which correspond to the original coefficient distribution for $n = 1$) are similar to the semi-circle as dimension increases.

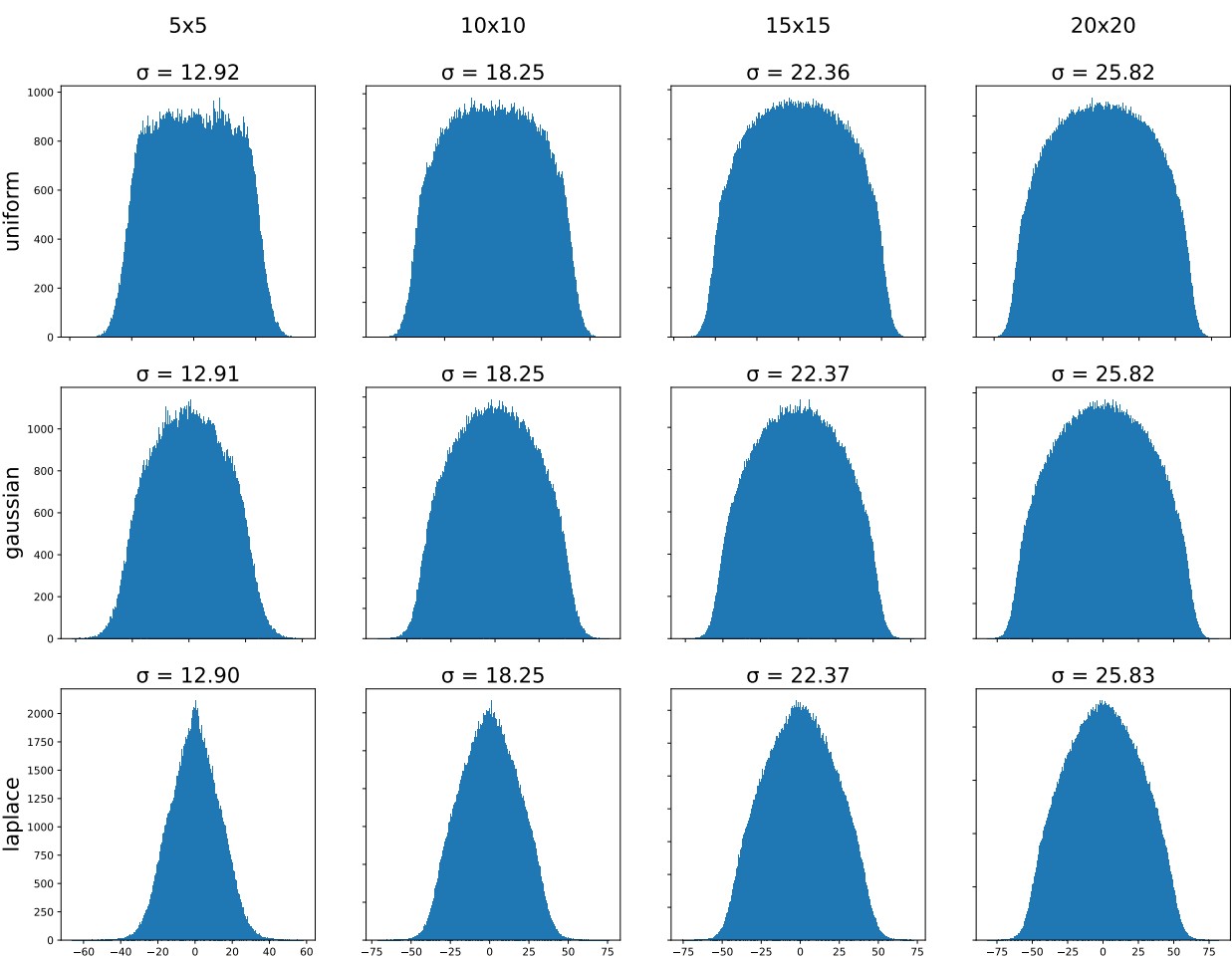

Figure 3: **Empirical distributions of eigenvalues for Wigner matrices**, dimension 5x5 (left) to 20x20 (right), with uniform (top), gaussian (middle) and Laplace (bottom) coefficients. All distributions computed from 100 000 random matrices.

