# OpenReview forum: "Linear algebra with transformers"
_TMLR — Accepted by TMLR_

### Review · Reviewer_XmY7 · 2022-08-05

**Summary Of Contributions:**

This paper investigates the numerical approximation of nine linear algebra problems, utilizing the Transformer.

For effective problem representation, four encoding schemes are proposed to represent real numbers in a matrix.

Random matrices and their corresponding solutions to each problem have been generated as the training data for Transformer. On small size (e.g. from 5$\times$5 to 20$\times$20) matrices, the Transformer architecture achieved high accuracy under the pre-defined error thresholds.

Moreover, noisy input and out-of-domain distributions are studied to show the robustness of the Transformer in dealing with generalization issues.

**Broader Impact Concerns:**

I do not have the concerns about this paper.

**Requested Changes:**

- More related work and a thorough comparison with existing works are encouraged.
- More baselines are suggested.
- The real value and potential application should be detailed.
- How about the inference time for different problems?

For more questions, please refer to the above weaknesses.

**Strengths And Weaknesses:**

## Strengths
- Nine linear algebra operations are studied in this paper, which covers a wide range of possible linear algebra operations.
- Four encoding schemes are proposed to represent the numerical real numbers, providing feasible solutions for the learnable numerical calculation problem.
- In general, the Transformer achieved high accuracy under certain error thresholds on the nine problems.

## Weaknesses
- Technically, the main contribution of this paper is the utilization of Transformer to solve the nine linear algebra problems. The application is somehow novel. But from the algorithm and model perspectives, there is not much contribution and new insights in this paper.
- In the main paper, there is no other baseline except for Transformer. For example, in Related Work, the authors stated: **"Algorithms using neural networks to compute eigenvalues and eigenvectors have been proposed since the early 1990s", "In a recent paper, Blalock & Guttag (2021) use learning techniques to refine known approximate algorithms for matrix multiplication."**. Basic LST, GRU, and a variant of Transformer (Universal Transformer) are experimented in the Supplimentary. But the line-to-line comparison is missing, which is not easy to compare.
- The Related work is too short. There are various related and previous works in dealing with mathematical, combinational problems using Deep neural networks, RNNs, etc. For example, the pointer network [R1] was proposed to solve the convex hull and TSP problems.
- I wonder about the potential value and usage of the proposed algorithm. Of course, some investigations and research tries are encouraged and need not necessarily surpass all existing algorithms/packages, if some insights can be given. In this paper, I did not see the real value of using Transformer to solve linear algebra problems, e.g. efficiency? large-scale capability? Moreover, all the experiments are with very small-scale matrices ($5\times 5$ to $20\times 20$). On such a small scale, one can even manually compute these results without any computer aids.
- For the more difficult SVD and eigenvector problem, this paper can only deal with the $5\times 5$ or $4\times 4$ scale, which is far from the real application. But the simple addition/transpose/multiplication only involves add/multiply operations, which are quite easy to solve and parallel in a deep learning era.
- For the precision, three significant digits in the mantissa are utilized. This is relatively small. Are there any other comparison methods/algorithms?
- How about the inference time? Is it comparable to or faster than existing numerical packages?





[R1] Vinyals, Oriol, Meire Fortunato, and Navdeep Jaitly. "Pointer networks." Advances in neural information processing systems 28 (2015).

---

> ### Author Response · Authors · 2022-08-26
> **Response to Reviewer XmY7 1/2**
>
> Thank you very much for your review.
>
> **From the algorithm and model perspectives, there is not much contribution and new insights in this paper.**
>
> The four number encodings are our main technical contribution. We propose additional innovations: asymmetric transformers and gated universal transformers. We added them to the appendix because we wanted the paper to focus on the problems and on our out-of-distribution (OOD) results. We can expand this part of the analysis and the corresponding discussion, and move it to the main section, if the reviewer deems it necessary.
>
> The other important contribution is our OOD results. Transformers are known to struggle with OOD generalization. In symbolic mathematics, this was mentioned by Lample and Charton (Deep Learning for Symbolic Mathematics, ICLR 2020) and further studied by Welleck et al (Symbolic brittleness in sequence models, AAAI 2022). The common interpretations is that transformers mostly memorize training examples and interpolate between them. This allows for in-domain generalization, but breaks once one steps out of the training distribution. As a result, many authors suggest to mitigate the problem by training on more diverse datasets, essentially turning OOD into in-domain generalization.
>
> In section 5.1 we show that a model trained on matrices with Laplace-distributed eigenvalues (i.e. positive and negative, symmetrically distributed around the origin) can generalize to positive definite matrices, a very different category. To our knowledge, this is the first result demonstrating this kind of OOD behavior. By carefully choosing the training distribution, generalization beyond interpolation of training data is possible. We will reinforce the corresponding discussion.
>
> **I wonder about the potential value and usage of the proposed algorithm.**
>
> We will extensively rewrite and expand the introduction to better explain our motivation. To clarify, we are not suggesting that transformers could, or should, be used as replacement for existing algorithms. Our goal in this paper is to explore whether transformers can be trained, from data only, to perform computations to high accuracy. That this is even possible is by no means obvious: previous papers (Kaiser & Sutzkever,  Neural GPUs Learn Algorithms, 2015, Palamas, Investigating the ability of neural networks to learn simple modular arithmetic, 2017, Nogueira et al, Investigating the limitations of transformers with simple arithmetic tasks, 2021) show that basic arithmetic can be difficult to learn. Researchers have even found that some problems of arithmetic are intractable with gradient descent (Shalev-Schwartz, Shamir and Shamma, Failures of gradient-based deep learning, PMLR 2017). On the other hand, transformers have successfully been trained on problems of symbolic mathematics, such as integration (Lample et al 2019).
>
> One direct motivation of our research is to understand the limitations of transformers, and how to cope with them. Understanding that problems of linear algebra, like computing eigenvalues, can be learned, while basic arithmetic cannot, is in our opinion an important finding. We also believe that our OOD results contribute to a better understanding of out-of-distribution generalization, an important direction of current research on transformers.
>
> Another practical motivation stems from the fact that transformers are becoming the “default model” for many applications, and are increasingly considered as end to end tools for AI for Science. In such applications, transformers must learn complex algorithms from data only. This will not be possible if transformers cannot perform common mathematical computations, like linear algebra.
>
> **How about the inference time? Is it comparable to or faster than existing numerical packages?**
>
> Eigenvalue calculation for a 5x5 matrices takes 0.5 millisecond with a 4/1 layer transformer (encoder/decoder), 1ms with 6/1 layers, and 1.5 ms with 6/6 layers. Matrix inversion takes about 1ms with a 6/1 layer transformer, running pyTorch on a single GPU machine.
>
> On the same machine, the optimized algorithms in Numpy (linalg.eigval, and linalg.inv) are faster : 0.07 millisecond for eigenvalues, and 0.04 ms for inversion. However, our code was not designed for speed. An optimized version of our models could perhaps achieve comparable speeds as those of numerical packages (but note that the memory footprint of a transformer would still be much larger). We will add these results in the discussion.
>
> But again, we are not advocating replacing existing packages with transformers: if one already knows that eigenvalues must be computed, or a matrix inverted, by all means use Numpy!

---

> > ### Author Response · Authors · 2022-08-26
> > **Response to Reviewer XmY7 2/2**
> >
> > **There is no other baseline except for Transformer.**
> >
> > We do provide comparisons with other neural architectures, and will complete and standardize these results (including a few more models). However, we would like to point out that our motivation for choosing transformers is not that they might perform better than other architectures one those specific tasks. We investigate transformers because they are quickly becoming the default solution for end to end modelling, and are increasingly considered for science. In our opinion, the important question is not whether transformers can do better than existing algorithms (they cannot), but whether they can handle the numerical tasks that will be needed to solve problems of science.
> >
> > Since we are working on problems which have known solutions, we do not quite understand the purpose of a baseline. All the problems studied here can be solved with 100% accuracy and for very large dimensions with the classical algorithms described in textbooks like Golub & Van Loan, or implemented in packages like Numpy.
> >
> > We disagree that the two papers you suggest could serve as baselines. The first one is a very different technique that solves eigenvalue problems at train time : the network must be retrained for every new test case, and there is no training phase. The paper by Blalock and Guttag covers the specific case where one matrix is reused in many different products: you want to compute A_1 B, A_2 B, … A_n B for large n. We will add discussion in the extended related works.
> >
> > **The Related work is too short.**
> >
> > We will expand the related works. Thank you for the references.
> >
> > **For the precision, three significant digits in the mantissa are utilized. This is relatively small. Are there any other comparison methods/algorithms?**
> >
> > Thank you for this observation. We originally chose 3 digit precision because it led to reasonable vocabulary size for the 4 encodings we studied (with three digits in the mantissa, FP15 uses about 30k words, with 4 digits it would run around 300k). Another reason to use moderate precision was that many algorithms exist that allow one to refine an initial approximation, at very little cost. If high precision was sought, such techniques could be used as a post-processing step.
> >
> > Following your suggestion, we experimented with different precision. We tried 2, 3 and 4 digit mantissas (with base 100, 1000 and 10000 positional encoding, i.e. 3 tokens per matrix coefficient) for eigenvalue calculations of 5x5 matrices (using the 4/1 layers, 512 dimensions and 8 head architecture from table 6).
> >
> > After 85 epochs (25.5 million examples) we achieve the following accuracies at 5, 1 and 0.5% tolerance:
> > * 2 digits, P100: 100 / 78.5 / 61.4
> > * 3 digits P1000: 100 / 91.6 / 50.0
> > * 4 digits P10000: 100 / 88.7 / 37.4
> >
> > For large tolerance levels (5%), precision seems to have little impact on performance. At higher tolerance, the use of a larger vocabulary seems to slow the learning when precision is higher, but comparable accuracies are eventually be achieved.
> >
> > We will add a section to the appendix, and provide experimental results for addition and eigenvalues.
> >
> > **For the more difficult SVD and eigenvector problem, this paper can only deal with the 5×5 or 4×4 scale, which is far from the real application. But the simple addition/transpose/multiplication only involves add/multiply operations, which are quite easy to solve and parallel in a deep learning era.**
> >
> > SVD is the only task that is limited to 4x4 and that fails to scale. We take that as a negative result (which shows that these problems are by no means trivial). Eigenvalue calculation, a nonlinear problem, scales to 20x20, and eigenvectors to 6x6 (but it is quite possible that the tricks we used to scale eigenvalue calculation, i.e. mixing problems of different size, could be applied to eigenvectors, we could add those experiments if you think they are necessary).
> >
> > In the paper, we chose to use very small models (less than 6 layers, BERT 2018, uses 12 and 24 layers, recent architectures are even deeper). It is possible that the apparent limitation to 5x5 matrices is due to model size and depth (there is a limit to the complexity of computations that can be performed with a given model depth). In fact, our experiments show a corrrelation between model size and problem complexity: we used
> >
> > - 1 layer transformers for transposition and small dimension addition
> > - 2 layers for large additions and matrix vector products
> > - 4 layers for matrix products and eigenvalues
> > - 6 layers for larger eigenvalues, eigenvectors and inversion
> >
> > We could experiment with larger models, if the reviewer think it is important.

---

> > > ### Comment · Reviewer_XmY7 · 2022-09-08
> > > **Thanks for the response, contributions and insights are insufficient**
> > >
> > > Thank you for the detailed comments and more experiments.
> > >
> > > Firstly, I agree with other reviewers that the problem is important and worth investigating, although it is far from a realistic application.
> > > Secondly, the authors claimed the main contribution is the four number encoding techniques. From my perspective, this technique is not sufficient for the TMLR bar.
> > > Thirdly, it is interesting that the transformer for matrix operations has some OOD properties, which shows potential future application. But by far, this is still far from more practical and realistic usage.
> > >
> > > In general, I agree with the improvement directions suggested by other reviewers. But I am still concerned about the technical contribution of this paper to TMLR.

---

### Review · Reviewer_zbpq · 2022-08-15

**Summary Of Contributions:**

The paper studies solving a collection of most common linear algebra problems using a classical transformer architecture.  While there are robust industry-strength algorithms for all of these problems that can solve them to numeric precision,  the goal here is not to compete with them, but instead to learn a rough approximation of the mapping from a gigantic collection of input-output example pairs: say given a collection of matrices and their inverses, the neural net should learn how to invert a new matrix from scratch.  The paper considers both fixed-size inputs and the harder case of variable size inputs (where the same architecture has to learn to invert 5x5 and 10x10 matrices), and uses collections of random (iid) matrices as training examples. On comprehensive collection of tests, it shows that for small examples (5x5 to 10x10 or so) indeed one can learn an approximate mapping after seeing hundreds of millions of paired examples, but the evidence seems to indicate that the network is mostly memoizing the neighborhood of the examples it has seen rather than learning much about how to really perform linear algebra.

**Broader Impact Concerns:**

It's important to stress/study the limitations of the proposed method -- as it has not ability to certify that its results are accurate or not.  I suspect that the choice of random matrices is strongly biasing the results, and may give over-confidence in the ability of the net to generalize.  e.g. if you give some specific non-random matrices (e.g. with high correlations or near-singularity) outside of your training distribution, the accuracy may be low -- and it's impossible to tell whether to trust the transformer or not.

**Requested Changes:**

I feel the paper would be much stronger if instead of asking "whether one can force transformers to do numerical linear algebra?" -- it would ask -- "what is the best neural architecture which can approximate linear algebraic functions?" -- and came up with something that has at least remotely conceivable practical use.  Giving some hint at specific problems where there's an unknown NLA-like function which one wants to learn from a large collection of input-output pairs would be helpful -- currently there's only a vague hint of possible applications.

Also trying to review/ apply existing system-identification approaches as a baseline would improve the scholarship in the paper, e.g.:
Ljung, Lennart. "System identification." Signal analysis and prediction. Birkhäuser, Boston, MA, 1998. 163-173.

In the current form the message that I learned from the paper -- is roughly that transformers are not a great fit to do NLA, and it takes enormous effort to make them show even a remote hint of learning in a highly restricted setting.  I think there's still value in the message -- "it's not a good idea to eat soup with a fork" -- but giving some analysis of why the problem structure is hard for transformers, and taking steps to address it would be significantly more valuable.

**Strengths And Weaknesses:**

Strengths:

* The paper provides a good example of the importance of having the right inductive bias -- and demonstrates that vanilla transformers are not the right tool for attempting to do numerical linear algebra (NLA), or even learning unknown linear-algebraic from pairs of examples.  It's a useful cautionary tale that there's no magic neural architecture which can solve all of the world's problems and considerable creativity is occasionally needed to design appropriate architectures to address new classes of problems.

* The paper conducts extensive numerical experiments, and the writing is generally clear.

Weaknesses:

* The paper makes minimal attempt to explain why transformers could be a good choice for solving linear algebra problems (apart from being powerful learners for language and vision problems), and uses standard transformers without trying to modify them or exploit any problem structure.  Why did you focus on transformers, and why do you think that densely-connected nets, or res-nets, or perhaps some wavenet-like architectures may not be a better fit?  The only notable attempt at adapting to the new problem is exploring 4 different representation of floating point numbers. For example, matrices are represented as a sequence of numbers, with the only hint to the neural-net about how to interpret them by providing two constants -- the number of rows and number of columns.  Do you have evidence that the transformer is indeed using these two variables successfully?  I suspect that keeping the input in matrix or tensor-form (akin to color-images) might do significantly better (and it's reasonable to expect that for problems that work on matrices the shape would be known a-priori).  For example keeping two matrices A and B stacked into MxNx2 tensor seems like it should have a far easier time to represent at least the addition operation.

* There's only a vague hint about the practical motivation for the problem.  There are many successful applications of neural nets which try to speed up a complex numerical computation, for example solving Navier-Stokes equations in air-flow calculations,  Jin, X., Cai, S., Li, H., & Karniadakis, G. E. (2021). NSFnets (Navier-Stokes flow nets): Physics-informed neural networks for the incompressible Navier-Stokes equations. Journal of Computational Physics, 426,   or using graph-neural nets to approximately solve combinatorial optimization problems:  Cappart, Quentin, Didier Chételat, Elias Khalil, Andrea Lodi, Christopher Morris, and Petar Veličković. "Combinatorial optimization and reasoning with graph neural networks." arXiv preprint arXiv:2102.09544 (2021).   In these approaches neural nets have been carefully designed to take advantage of the problem structure (engineer an appropriate inductive bias) -- rather than applying a generic architecture designed for an unrelated task.   I suspect that using custom-made layers -- something akin to OptNet (Amos, B., & Kolter, J. Z. (2017, July). Optnet: Differentiable optimization as a layer in neural networks. In International Conference on Machine Learning (pp. 136-145). ) -- would be needed to provide a convincing attempt at learning NLA (numerical linear algebra) functions.

* What is the goal here?  Even if the performance was much better (~90% accuracy for linear algebra restricted to specific random matrices of miniscule size is utterly underwhelming -- useful NLA algorithms are accurate to close-to-numeric precision), I still don't understand how it could be used in practice.   Multiplying,  inverting or eigen-value decomposition of 5x5 or 10x10 matrices takes from 1 micro-second to about 20 micro-seconds on a generic cpu.  So even ignoring the multiple days of intense model training on several GPUs, at test-time feeding the matrices through the transformer to make predictions would not be faster.  If the goal is to learn an unknown complex function which may involve multiple linear-algebraic operations -- then there's a mature field called 'system identification', which has a collection of tools, which should be studied / mentioned / compared too.  For example, there are clearly much more effective tools to learn linear and bi-linear functions from data than feeding them through a transformer.  I think building a custom neural net which may borrow some modules from that literature may provide a dramatically better solution.  Also you should be more specific of what kind of phenomena you're trying to learn? -- maybe give some specific examples (linear algebra at large is too vague)...  There's a reason linear systems are so prevalent in science and engineering -- akin to gaussian distributions -- they indeed can serve as a very accurate description for a variety of natural phenomena -- hence why linear algebra is so widely used.

* The choice of random matrices as a test-suite could be very problematic -- random matrices are a very special sub-family of matrices, for example they exhibit strong concentration of measure (although it's probably more relevant to dimensions higher than 5x5).  For example, columns of random matrices will be approximate orthogonal, and will have a certain norm.  So it's not clear if the net is learning something about the linear-algebraic task, or some property pertaining to the specific random matrix ensemble.  It would be very interesting to see how the learned net performs on a generic collection of matrices arising in real applications, or in common test-suites for linear-algebra -- for example perhaps something like the matrix market collection from NIST (US national institute of standards and technology).

* Given 100's of millions of examples of 5x5 matrices,  it seems like you may be pretty well covering the space of such matrices -- so you may simply be memoizing local neighborhoods rather than really learning how to do NLA.  How would something like a basic K-NN method compare with a similar number of examples?  Alexey Efros presented some evidence that simple KNNs with gigantic training collections can sometimes compare to state-of-the art neural nets on some vision and annotation tasks (Neurips 2017 presentation on unreasonable effectiveness of nearest neighbors).  Are you doing something significantly more effective than simply memorizing neighborhoods of these 100's of millions of examples? The difficulty that you have with applying a single transformer to learn NLA with matrices of different size -- suggests that it's likely not learning much about specific NLA operations...

---

> ### Author Response · Authors · 2022-08-26
> **Response to Reviewer zbpq 1/2**
>
> Thank you for your thoughtful comments. We will first address our motivation and goals, which features in several of your questions. We will extensively rewrite the introduction of our paper, in order to better explain our motivation.
>
> **What is the goal here?**
>
> We do not advocate replacing existing linear algebra algorithms with transformer-based implementations. We agree with your observation about having the right inductive bias: if one knows that a linear algebra operation is to be performed, numerical packages are the way to go (at least for the small and dense matrices considered here), and there is little need for replacements, neural or else.
>
> Our motivation is twofold. We want to better understand the capabilities and limitations of transformers in mathematics, and their potential use as tools for the emerging field of AI for science. It has been known for a while that sequence models, and especially transformers, struggle with simple arithmetic tasks Kaiser & Sutzkever,  Neural GPUs Learn Algorithms  2015, Palamas, Investigating the ability of neural networks to learn simple modular arithmetic, 2017, Nogueira et al 2021). It could even be proven that some problems of arithmetic are intractable for gradient-based techniques (Shalev-Schwart, Shamir and Shamma, Failures of gradient-based deep learning, PMLR 2017). On the other hand, transformers were successfully used for other mathematical tasks, like symbolic integration and theorem proving. On many of these problems, transformers have been shown to struggle with out of distribution (OOD) generalization (Lample and Charton, Deep Learning for Symbolic Mathematics, ICLR 2020, Welleck et al (Symbolic brittleness in sequence models, AAAI 2022). Our work suggests that whereas transformers are limited in arithmetic, they can perform advanced (but approximate) computations.
>
> A more practical motivation is provided by the increasing interest in transformers to solve problems of science. We believe this will only be possible if transformers can handle the computations that form the “basic building blocks” of scientific applications. Linear algebra is one of those. Demonstrating that transformers can solve problems of linear algebra (even at a lesser level than dedicated algorithms) is a prerequisite to their generalization.
>
> **Why did you focus on transformers, and why do you think that densely-connected nets, or res-nets, or perhaps some wavenet-like architectures may not be a better fit?**
>
> We provide comparisons with different architectures in section D of the appendix and will add some more, but our main reason for focusing on transformers is that they are becoming the default model for many ML applications, including AI for science. Our main goal, in this paper, is to explore their limitations and their capability to learn basic mathematical operations, like linear algebra. That this is possible is not obvious: recurrent networks and transformers have a lot of difficulty to learn such basic operations are integer multiplications. On the other hand, if basic and ubiquitous calculations (like operations on matrices) cannot be performed by transformers, their use as models for AI for science is severely limited.
>
> **Do you have evidence that the transformer is indeed using these two variables successfully?**
>
> In the transposition experiments with rectangular matrices of variable size, we learn to transpose matrices with different numbers of rows and columns. To transpose matrices with the same number of coefficients, but different shapes (e.g. 5x12, 6x10, 10x6 and 12x5), the model must learn different permutations of the coefficients, and the only way to tell one from the other is by the number of row and column tokens. Our model achieves 99.6% accuracy on this task, which would not be possible if the variables were not used.
>
> **I suspect that keeping the input in matrix or tensor-form (akin to color-images) might do significantly better**
>
> This is essentially what the FP15 encoding is doing: each coefficient is encoded as a single token, which is then embedded (via a transfer matrix of learnable parameters) in high dimensional representation space. Our experiments show that this is not always the best choice. In particular,  multiplication is easier to learn with the three token representation (sign, mantissa, exponent), probably because it provides better inductive bias about the basic operation: to multiply, you sum exponents, multiply, truncate and carry mantissas, and perform 2-element multiplication on the signs, the three token encoding makes this easier to learn.
>
> **I suspect that using custom-made layers (...) would be needed to provide a convincing attempt at learning NLA (numerical linear algebra) functions.**
>
> Such models would certainly learn faster, but if the problem to be solved (linear algebra) is already known, what is the point of using a neural net, instead of numerical packages that provide all the necessary guarantees?

---

> > ### Author Response · Authors · 2022-08-26
> > **Response to reviewer zbpq 2/2**
> >
> > **The choice of random matrices as a test-suite could be very problematic -- random matrices are a very special sub-family of matrices, for example they exhibit strong concentration of measure (although it's probably more relevant to dimensions higher than 5x5). For example, columns of random matrices will be approximate orthogonal, and will have a certain norm.**
> >
> > For dimension 5, with the generator we use (uniform between -10 and 10), the cosine of the angle between two columns has standard deviation 0.44 (and zero average), for dimension 10, standard deviation is 0.31. This is not approximately orthogonal. Samely, norm has average 12.6 and standard deviation 2.7 for dimension 5, and 18 and 2.6 for dimension 10.
> >
> > The main concentration law that applies here is the semi-circle law. Its impact is discussed in section 5 (OOD). Section C.3 of the appendix also demonstrates that models can be trained (for eigenvalue computation) on different ensembles, to similar accuracy. We could extend this to other tasks if the reviewer deems it important.
> >
> > Testing on “natural” examples of matrices is an interesting idea. However, the matrix market collection seems mostly focused on large sparse matrix, which are not considered in this paper. In fact the only set that seems to produce small dense matrices is XLATMR, which uses the same random generators as us.
> >
> > **Given 100's of millions of examples of 5x5 matrices, it seems like you may be pretty well covering the space of such matrices -- so you may simply be memoizing local neighborhoods rather than really learning how to do NLA. (...) The difficulty that you have with applying a single transformer to learn NLA with matrices of different size -- suggests that it's likely not learning much about specific NLA operations...**
> >
> > Our training samples are actually much smaller. Section C.2 in the appendix provides training speeds for different models, and shows that our best models for addition and multiplication need less than 2 million examples to reach 95% accuracy. For eigenvalues, the training size is less than 4 million examples. We often push these numbers to achieve higher accuracy at high tolerance, but accuracies over 90% can usually be achieved after just a few million examples.
> >
> > Three observations make us believe that memoization and interpolation do not account for most of our model accuracy:
> >
> > * In section 4.4, we show that the eigenvalues of matrices with more than 8 dimensions are very difficult to learn from a training set of matrices with fixed dimension. On the other hand, training from matrices with different sizes allows our model to learn eigenvalues for dimensions up to 20x20. This suggests that some common properties of matrices of different sizes are leveraged by the model (memoization would not work for matrices of different dimensions).
> >
> > * In section E.1, we show that it retraining a model on matrices of different sizes takes significantly less examples than train it from scratch. If memoization was at work, there should be little benefit to retraining.
> >
> > * Most importantly, our results on out of domain generalization seem to rule out interpolation: a model trained on matrices with Laplace distributed eigenvalues (which can be positive or negative) will generalize to positive definite matrices, a completely different ensemble.
> >
> > **Then there's a mature field called 'system identification', which has a collection of tools, which should be studied / mentioned / compared too.**
> >
> > We agree that a comparison between black-box tools for system identification and transformers (or similar architectures) would be extremely interesting, but we feel this is the subject of a different paper. We will mention and discuss system identification in the related works.
> >
> > **It's important to stress/study the limitations of the proposed method -- as it has not ability to certify that its results are accurate or not.**
> >
> > We will add a section about it in the discussion. In our opinion, our work has two main limitations: the dimension of matrices is limited by the size of the sequences a transformer can process (a few thousand tokens in the architecture we use), and we only consider dense matrices.
> >
> > On verification, the model cannot predict whether a solution is correct or not, but external validation is sometimes possible. An inverse, computed by the model could be multiplied by the original matrix, and compared to identity. This means using an external verifier, but it is a simpler operation than inversion. Similar techniques could be applied to SVD and eigenvalue decomposition.

---

> > > ### Comment · Reviewer_zbpq · 2022-08-30
> > > **Thank you for the response, more insight is needed.**
> > >
> > > Thank you for the detailed comments.  I generally agree that understanding the power and limitations of transformers is an interesting and valuable direction of research (even if it may not lead to immediate applications). I agree that evidence suggests that transformers  are able to go beyond simple memoization, and seem to exploit some partial structure in linear-algebraic problems. However, they're still very far from usefully "solving" these problems.  Unfortunately the paper does not give much insight into what are they learning / what structure they're exploiting, and where are they falling short -- and so provides no visible pathway to try to improve/modify them to make better progress on this class of problems. I agree with reviewer cm5p in terms of directions to improve the paper.

---

### Review · Reviewer_cm5p · 2022-08-21

**Summary Of Contributions:**

The paper studies how well transformer can solve linear algebra problems. The problems considered include basic matrix operations (addition, multiplication, transpose) to eigenvalue decomposition and inversion. The paper uses dense transformers and mostly train a specific network for each problem. The problem size is limited to small matrices. The main finding is that transformers can perform well on these tasks, up to a small pre-set error threshold (tolerance). The paper also provides specific results on robustness to noise and out-of-distribution generalization. For OOD, it shows that models trained to predict Laplace-distributed eigenvalues generalize to other classes of matrices such as Wigner matrices or matrices with positive eigenvalues.



**Requested Changes:**

Please address weaknesses pointed out in the previous section of my review.

Note that co-training is a standard term in ML. Quoting wikipedia, "co-training is a semi-supervised learning technique that requires two views of the data. It assumes that each example is described using two different sets of features that provide complementary information about the instance." Please replace co-training with joint training.

**Strengths And Weaknesses:**

Strengths

1. The paper presents a comprehensive set of experiments on solving linear algebra problems using transformers.

2. It proposes four different encodings of input matrices.

3. The paper also shows the models are robust to noise (random errors of matrix entries).

4. The paper presents a specific result on out-of-distribution generalization, namely models trained to predict Laplace-distributed eigenvalues generalize to other classes of matrices such as Wigner matrices or matrices with positive eigenvalues.

Weaknesses

1. The results are limited to very small matrix sizes. Can the model generalize to matrix sizes not trained? For example, there has been techniques since 2015:

[1] Making Neural Programming Architectures Generalize via Recursion

Jonathon Cai, Richard Shin, Dawn Song

ICLR 2017

[2] Neural GPUs Learn Algorithms

Łukasz Kaiser, Ilya Sutskever

2015

2. Theoretically all problems of such small sizes should be solved with 100% accuracy with transformers. Can the authors train with more data or better synthetic training data to achieve that?

3. The results are just reported as they are without any deeper insights. For those not achieving 100% accuracy currently, what errors does the model make? It was known that models take short cuts. For example, not able to learn general algorithms to solving addition or multiplication, but learns specific algorithms limited to input length. What exactly are transformers learning? I would encourage the authors provide deep insights as done in other papers, e.g.

[3] Transformers discover an elementary calculation system exploiting local attention and grid-like problem representation

Samuel Cognolato, Alberto Testolin

2022

4. The results presented on alternative model architectures, LSTM and GRU also lack insights. What makes transformers learn better than LSTM and GRU? I believe if LSTM and GRU are properly designed and tuned on hyperparameters, and with the right training data, one can achieve 100% accuracy.

---

> ### Author Response · Authors · 2022-08-26
> **Response to Reviewer cm5p**
>
> We thank the reviewer for their comments and questions.
>
> **The results are limited to very small matrix sizes. Can the model generalize to matrix sizes not trained?**
>
> Our models cannot generalize to dimensions not seen at training (see Section E1 in the appendix) because the corresponding sequence lengths were never seen. This is a known limitation of transformers, generally attributed to positional encodings.
>
> We agree that generalizing to yet unseen dimensions would be a nice addition. In fact, this was our original motivation for using universal transformers (section D.2), as they make transformers recurrent (as in the references you provided), and recent papers have shown that this can allow extrapolation to large dimensions (Schwarzchild et al, can you learn an algorithm? Neurips 2021, Bansal et al, End-to-end Algorithm Synthesis with Recurrent Networks, 2022). However, our initial results were disappointing. We believe that relative positional embeddings must also be introduced. We see this as a potential follow-up, but we could add initial results if you think this is important.
>
> On this topic, the paper includes two related results:
>
> 1- that mixing matrices of different dimensions can help learn larger problems: in section 4.4, we show that whereas the eigenvalues of 10x10 matrices are almost impossible to learn when one trains the model on 10x10 matrices only, we can learn to compute all eigenvalues up to 20x20 by mixing dimensions in the training set.
>
> 2- that re-training a model on matrices of different dimensions needs less training data than training from scratch. We can further investigate this phenomenon if the reviewer wants, notably by looking at the potential role of pretraining on a large set of matrix operations and dimensions.
>
> We will add a discussion of these results in section 7.
>
> **Theoretically all problems of such small sizes should be solved with 100% accuracy with transformers. Can the authors train with more data or better synthetic training data to achieve that?**
>
> On 5x5 matrices, we achieve 100% accuracy for all tasks except matrix inversion, eigenvectors and SVD. For inversion, the 90% limit seems very hard to break with our current architectures. Two solutions are possible : try larger models, and add a little more diversity to our training data, either in the form of Wigner matrices with different coefficient range, or matrices with Laplace distributed eigenvalues (since they help with generalization). We could add experiments to this effect.
>
> **The results are just reported as they are without any deeper insights. For those not achieving 100% accuracy currently, what errors does the model make?**
>
> Our main insight on failure cases can be found in section B2, figure 1, and section C1 figure 2 of the appendix. Section B2 shows that, for all problems, when accuracy is measured with L2 norm (overweighting the largest errors), models converge to 100%, even when L1 norm does not (eigenvalues 10x10). Section C1 indicates that accuracy to larger tolerance is reached faster. Overall, this seems to suggest that models learn better and better approximation of the solutions, instead of “breaking down” in some specific cases.
>
> We will conduct an additional analysis of the failure cases for matrix inversion, to measure the proportion of failure cases due to “imperfect approximation” (e.g. 7% relative L1 error instead of the 5% tolerance), and the nature (ie distribution of eigenvalues) of the hard to approximate matrices.
>
> We can provide attention visualization and t-sne plots of the embeddings. They do look nice, and show that the model “is learning something”, but they are notoriously difficult to interpret, as soon as models have more than one layer (which is always the case here, since we use sequence to sequence architecture).
>
> **The results presented on alternative model architectures, LSTM and GRU also lack insights. What makes transformers learn better than LSTM and GRU? I believe if LSTM and GRU are properly designed and tuned on hyperparameters, and with the right training data, one can achieve 100% accuracy.**
>
> Our results from tables 16 and 17 indeed show that proper hyperparameter selection is very important for LSTM and GRU. In particular, deeper architectures seem harder to train. Our results confirm the common observation that transformers are more flexible than LSTM and GRU, and can cover a broader range of problems. But the fact that some of our problems could be solved to high accuracy by LSTM came as a surprise, since most previous attempts to use them for mathematics had proven disappointing.
>
> **Please replace co-training with joint training.**
>
> We will rename the section and change the words, thank you for this comment.

---

> > ### Comment · Reviewer_cm5p · 2022-08-27
> > **Thank you for your clarification; need more contribution**
> >
> > I appreciate the authors working on an interesting and challenging problem. The current paper does not have enough contribution for publication at TMLR. I encourage the authors to make progress as follows.
> >
> > 1. Make meaningful progress on models that can generalize to matrix sizes not trained.
> > 2. Achieve almost 100% accuracy for linear algebra problems of small sizes for transformer as well as LSTM and GRU.
> > 3. Provide insights on when transformer is better than LSTM and GRU.
> > 4. Provide insights on when transformer makes mistakes.

---

### Author Response · Authors · 2022-09-01
**Additional results on failure modes 1/2**

Following reviewer comments, we experimented with failure modes for two experiments : matrix inversion and eigenvector computation (on other tasks, such as basic matrix operations and eigenvalue computation, our models train to 100% accuracy). These results will be added to the edited version of the paper (due Saturday the 3rd).

For each experiment, we use a trained model: the 6/1 layer, 12/8 heads, FP15/P1000 model from table 8 for inversion (90% accuracy), and the 6/1 layers FP15/P1000 model from table 7 for eigenvectors (93% accuracy). We generate a new test sample of 10 000 problems, have the models predict solutions, and evaluate performance using different metrics.

Our goal is twofold :
- characterize failure cases: understand which matrices are hard to invert or decompose
- provide insight on what the model does: which subtasks are easy, or hard

**Inversion**

Model accuracy on the test set is 89.6% at 5% tolerance. Accuracy drops to 81.7, 59.1 and 23.7 at 2,1 and 0.5% tolerance. This is consistent with the results in table 8 (computed from a different test set).

We first note note that all 10000 model predictions are well-formed matrices: there are no meaningless output, like incorrect encoding of numbers, or matrices with the wrong number of elements. This was consistently observed in all experiments: output syntax is learned to near-perfection at the beginning of training. We also note that accuracy increases for larger tolerance: we have 95% accuracy at 15% tolerance, and 96.6% at 25%. These two observations suggest that when our models fail, they do not hallucinate irrelevant solutions (as often happens in natural language), but still provide some (bad) approximation to the correct solution.

As stated in section 3, to evaluate our prediction, we compute PI, where I is the input matrix, and P the predicted inverse, and measure how far it is from identity (in L1 distance). This is the correct way to assess that an inverse has indeed been found. However, during training, the model is tasked with predicting the (unique) inverse of the input matrix. Therefore, another, weaker, metric is the L1 distance between the predicted inverse and the correct one (rounded to the same precision). On this metric, we achieve 98.2% accuracy at 5% tolerance, and 96.0, 92.3 and 84.5% at 2, 1 and 0.5 tolerance. In this metric, almost all failure cases are bad approximations: we achieve 99.3 accuracy at 15% tolerance and 99.5 at 25%.

This suggests that most of model failures happen when the (good) approximation of the inverse is not a "good inverse" of the input matrix, in the sense that PI is not close to identity. Theory tells us this happens when the condition number of the input matrix (the ratio of largest to smallest singular values) is large. Indeed, 98% of our correct predictions (5% tolerance) correspond to matrices with condition number below 51.5. On the other hand, 98% of failures are matrices with condition numbers larger than 51.5. The condition number of the input matrix proves to be a very accurate predictor of model success.

These results provide a complete explanation for model failures on this task. They indicate that model failures are not due to problems with the architecture or learning technique, but to the mathematical limitations of the approximate computation of matrix inverses (which apply to every numerical algorithm). They also indicate that failures happen on a definite class of problems, and can be predicted in advance.

They also suggest two directions for improvement. First, we could oversample ill-conditioned matrices in the training set, in a form of curriculum learning. Second, since ill-conditioning amplifies the effect of rounding and approximate computations, training with increased precision should improve accuracy. Other stabilizing tricks, which imply modifying the input matrix to lower its condition number (e.g. for symmetric matrices, adding a small diagonal) could also be applied.

---

> ### Author Response · Authors · 2022-09-01
> **Additional results on failure modes 2/2**
>
> **Eigenvectors**
>
> On the new test set, the model achieves 91.1% accuracy with 5% tolerance, and  82.1, 45.2 and 1.4% at 2, 1 and 0.5 tolerance. We predict two matrices, the diagonal matrix of eigenvalues D and the orthogonal matrix of eigenvectors H, and measure accuracy using the L1 distance between H' I H (I the input matrix, prime denotes transposition) and D. That is, we test the capability of H to diagonalize the input matrix into D.
>
> As in the inversion experiment, almost all model predictions are syntactically correct (only one case in 10000, here), and many errors are in fact "worse approximations" (95.6 accuracy at 25% tolerance).
>
> Since this problem does not have a unique solution (eigenvectors are defined up to low dimensional rotations and symmetries, depending on the order of eigenvalues), measuring the distance between the predicted solution (D and H) and the solution from the test set fails: 14.1% accuracy at 5% tolerance. But instead of measuring the quality of the diagonalization (the distance between H' I J and D), we could measure the quality of the reconstruction (the difference between H D H' and I). This results in slightly better accuracies: 95.7% at 5% tolerance, 92.9, 88.4 and 73.0 at 2,1 and 0.5%. This metric corresponds to a weaker problem: finding approximations of I of the form H D H'.
>
> Theory tells us that D should correspond to the eigenvalues of the input matrices, and that H should be orthogonal (all columns orthogonal to each other, and with unit norm). We now use these criteria as additional metrics, to try to understand which operations the model is performing.
>
> We first note that eigenvalues are correctly predicted in all test examples: accuracies are 100, 99.9, 99.9 and 99.4 with 5, 2, 1 and 0.5% tolerance. This easier sub-task is learned by the model. The norms of the columns of H (all equal to one according to theory) are within 5% of 1 in 99.9% of the examples (and within 2, 1 and 0.5% in 99.7, 99.2, and 98.0%). All predicted eigenvectors are unitary. Finally, the eigenvectors should be orthogonal (and H should have condition number 1). On the test set, we find that the dot products of successive eigenvectors, which should be 0, are all within -0.05 and 0.05 in 93.6% of the cases (and within -0.01 and 0.01 in 85.1%). In other words, the model learns the eigenvalues, and predicts unit vectors as the eigenvectors, but sometimes fails to provide a strictly orthogonal matrix.
> This provides us with a criterion for assessing the quality of the model prediction: the condition number of the predicted H. 98% of our correct predictions (with 5% tolerance) predict a H condition number smaller than 1.035, and 98% of failures have a condition number larger than 1.04.
>
> These results provide the following insights.
>
> 1- when trained on a complex task like eigen decomposition, the model learns the easier sub-task of predicting eigenvalues with 100% accuracy. It also learns to preserve theoretical properties of the result, like the unit norm of eigenvectors, and their (approximate) orthogonality.
>
> 2- all failures seem to concentrate on specific input classes (ill-conditioned matrices for inversion) or sub-tasks (orthogonalizing eigenvectors for eigen decomposition). This allows us to develop accurate predictors of model failure: input conditioning for inversion, predicted H conditioning for decomposition.

---

### Author Response · Authors · 2022-09-03
**Revised version of the paper**

We have uploaded a revised version of the paper. Major changes are:

- the introduction was rewritten, to better explain our motivation and take reviewer feedback into account
- analysis of the failure cases for the eigenvector and inversion experiments were added to subsections 4.5 and 4.6 (for the main findings, see our comments posted a few days ago on the forum)
- the related work was significantly expanded (it doubled in size)
- the discussion was expanded, in particular we added a paragraph on memoization, one on comparisons with Numpy (discussing inference speed), and one on the verification of model results
- we added a discussion about precision in section C of the Appendix
- as suggested, we renamed "co-training" as "joint training"

Scaling experiments are running, but will take a few weeks to complete. We have been experimenting with models between 40 and 800 million parameters. Specifically, we are asymmetric models with 6, 8, 12 and 24 layers in the encoder or decoder, scaled as BERT (512, 640, 768 and 1024 dimensions, 8, 10, 12 and 16 heads), together with versions with larger dimensions, and versions with more heads, on the eigenvalue, eigenvector and inversion task.

Current results, suggest that larger models learn to higher accuracy, with more precision, and with less examples. For eigenvalues of 8x8 matrices, we reach 99% accuracy at 0.5 tolerance (vs 31% in the paper), for 10x10 matrices, we achieve 100% at 5% tolerance (and 96% at 2), vs 25% (and 0.4) in the paper. Experiments on eigenvectors and inversion are still running, but it seems likely that close to 100% accuracies can be achieved (as requested by a reviewer) just by scaling the model.

We thank again the reviewers for their constructive suggestions, which help improve the quality of the paper.

---

### Author Response · Authors · 2022-10-28
**Camera ready version**

Compared to the post-review updated version, the camera ready version includes the following changes:

- small style clarifications and typos throughout the paper
- use of pronoun "I" instead of "we"
- added a paragraph in the main (section 4.4) and a section in the appendix (D.3), with experiments on large models for the eigenvalue task; these experiments were launched during the review, but took too long to be incorporated in the previous version
- added the address of the source code repository (it should be available in a few days)

I would like to thank the reviewers and editors for their comments, they really made this paper better!

---

### Decision · Action_Editors · 2022-10-05

**Recommendation:** Accept as is

**Comment:**

Dear Authors and Reviewers,

Thank you all for the hard work and for having engaged in the discussions. The outcome of the discussions lead all three reviewers to recommend rejection. Nevertheless, I have decided to recommend to ACCEPT the paper, and below I explain the reasons for that decision that goes against the reviewers consensus.

I recall the TMLR Editorial Policies found here: https://jmlr.org/tmlr/editorial-policies.html The present paper adresses a problem related to the third point in the TMLR scope: "accounts of applications of existing techniques that shed light on the strengths and weaknesses of the methods"

In the TMLR guidelines, it is explained that the acceptance or not of a paper is set by the answer to two main questions:

    Are the claims made in the submission supported by accurate, convincing and clear evidence? I believe the numerical experiments are extensive and convincing.

    Would some individuals in TMLR's audience be interested in the findings of this paper? I believe so, as some of the reviewers also pointed out)

In the present case, despite the paper does not give an answer nor true insights to the "why" (as raised by all reviewers and I agree with them), it nevertheless convincingly shows that Transformers are not at the moment really appropriate for linear algebra tasks; so this paper "accounts applications of existing techniques that shed light on weaknesses of the methods" Indeed, despite some encouraging results on their learning capabilities, I agree with the reviewers that the sizes of the instances studied here and the number of training examples needed to reach results that are far from being on-par with standard linear algebra packages, lead me to concluding that Transformers are not competitive. Concerning the comments by some reviewers on trying to achieve almost 100% accuracy for linear algebra problems of small sizes: the fact that this is not the case despite the large data sets employed here is in itself a relevant observation, potentially surprising given the power of Transformers on some other seemingly more complex tasks, so I do not see that as a negative point of the paper. So from a certain perspective these are results in themselves backed up by extensive numerical tests (criteria 1) that may trigger interest in some of the TMLR audience (criteria 2) on "why is that the case?" and "how to improve this?". So overall I believe that in spite of the valid and meaningful concerns raised by the reviewer, both TMLR criteria are met and so the paper deserves publication.


**Audience:**

Part of the TMLR audience will be interested by the results. I have in mind mostly applied researchers working on the use of NN for scientific applications, and people that would be keen in getting more insights on why the current Transformers architectures are not great for simple tasks such as those studied here. Given the excitement going on around Transformers these days, I have no doubt that the paper will trigger some interest (and doubts).

**Claims And Evidence:**

The present paper showcases the power (and mostly limitations) of certain Transformers architectures for basic linear algebra problems. The numerical tests are extensive and convincing.